# Intelligent Novel IMF D-SWARA—Rough MARCOS Algorithm for Selection Construction Machinery for Sustainable Construction of Road Infrastructure

Bojan Matić [1], Milan Marinković [1], Stanislav Jovanović [1], Siniša Sremac [1] and Željko Stević [2,*]

[1] Faculty of Technical Sciences, University of Novi Sad, Trg Dositeja Obradovića 6, 21000 Novi Sad, Serbia; bojanm@uns.ac.rs (B.M.); milan.marinkovic@uns.ac.rs (M.M.); stasha@uns.ac.rs (S.J.); sremacs@uns.ac.rs (S.S.)

[2] Faculty of Transport and Traffic Engineering, University of East Sarajevo, Vojvode Mišića 52, 74000 Doboj, Bosnia and Herzegovina

\* Correspondence: zeljko.stevic@sf.ues.rs.ba or zeljkostevic88@yahoo.com

**Abstract:** The quality of road infrastructure largely depends on the quality of road construction and adequate construction machinery. In order to reduce uncertainties and improve the performance of road infrastructure, it is necessary to apply modern and appropriate construction machinery. The aim of this study was to create a novel integrated multi-criteria decision-making (MCDM) model for the selection of pavers for the middle category of roads. A total of 16 criteria were defined and then divided into four main groups, on the basis of which the performance of 12 pavers was evaluated. Improved Fuzzy Stepwise Weight Assessment Ratio Analysis (IMF SWARA) with D numbers (IMF D-SWARA) was extended to determine the significance of the criteria for the selection of construction machinery based on two groups of experts. Rough measurement of choices and their ranking as a compromise solution (R-MARCOS) was used to evaluate and rank the performance of construction machinery. The results show that three alternatives out of the set considered can satisfy defined requirements. After that, we performed a multi-phase validity test in which different values of criterion weights were simulated. A comparative analysis with seven other Rough MCDM methods was also created, and the Spearman's correlation coefficient (SCC) and WS coefficient were calculated to determine the correlation of ranks for sensitivity analysis and comparative analysis. Thus, the obtained results were verified.

**Keywords:** road infrastructure; construction; D numbers; IMF SWARA; Rough MARCOS; construction mechanization



## 1. Introduction

The complexity of construction projects is constantly increasing under the influence of accelerated economic, political, and social changes; increased demands for energy efficiency and resource usage efficiency; and the accelerated development of new technologies. Consequently, efficient decision-making in the construction industry is of great importance in order to meet all functional, technical, economic, and even environmental project goals. The development of road transport is constantly increasing, both in terms of the number of vehicles and in relation to the height of axle pressures. As the cost of transport per unit of weight decreases with increasing vehicle speed, the tendency to enable faster and safer vehicle movement requires the use of state-of-the-art equipment when constructing roads. Currie [1] states that efficient execution of projects in the civil engineering sector is best ensured by the introduction and application of modern mechanization, which must be carefully selected in accordance with the requirements and needs of a particular project. An important part of such mechanization is a paver, which is a complex self-propelled machine on a stand with caterpillars or rubber wheels for making load-bearing layers of asphalt–concrete mixtures for roads, airfields, canal linings, etc. The paver consists of components

that take, spread, and compact the mass, and if necessary, they can serve as smoothers or cut the built-in surfacing. There are many different types of pavers in terms of their size, practical performance, and moving construction, as well as gripping and installation of materials, and this is the reason for evaluating their performance and selecting the best one. The advantages of the paver are that the precision levels of finishing are very high, the entire operations are mechanized, it is suitable for a large paving area, and it allows for automatic width adjustment without unnecessary adjustments [2]. The paver is an important road construction machine, as it saves labor costs and increases efficiency.

In order to make a proper selection of pavers, in addition to the capacity of the paver, i.e., the capacity of mass installation and various characteristics of the machine itself, other factors should be taken into account, such as the type of work to be performed, the strength required to complete the tasks in a given time, etc. Therefore, for the realization of defined jobs, it is necessary that the installation speed be as high as possible and the installation time to be kept at a minimum. However, by selecting a paver with an appropriate installation speed, but with too much weight (too much weight affects the reduction of flexibility, mobility, and maneuverability), there is a risk of getting the opposite effect. This is an additional reason for the proper evaluation of their performance that is achieved by applying the developed intelligent hybrid MCDM model.

One of the most important approaches to reducing costs in the construction sector is presented through assessing the performance of different types and subcategories of construction machines in different operating conditions and considering critical machine performance (engine speed, engine type, operating hours, torque or engine power, machine weight, type of fuel, and service life of equipment) [3].

Sinenko et al. [4] presented the calculations that need to be used when selecting construction machines and determining their number. Economic indicators of construction production are directly related to the selection of the optimal number of machines. The lack of machines for construction and installation works leads to the impossibility of their implementation in the directive period. The pace of construction and installation work and labor productivity largely depend on the degree of provision of construction facilities with machines. When determining the need for machines for construction sites, it is necessary to take into account the provision of work within the planned time, increase the level of complex mechanization, increase work productivity, and reduce manual work.

Given that it has been proven that, through an adequate assessment of the performance of construction machinery, costs can be reduced in the field of construction. The main goal of the study was an intelligent analysis of the decision-making process. The main goal can be manifested through the model for evaluation and selection of construction machinery in order to ensure sustainable construction of road infrastructure. This analysis should be useful for designers, planners, and other decision makers because it takes into account a set of important factors that fall into a group of economic, environmental, and technical and technological criteria. It is important to note that, in such analyses, there are conflicting criteria that increase uncertainties in decision-making, and these can be overcome through the development of a model such as the one presented in this paper. Based on the literature review and consultations with experts in the field of civil engineering, we defined a set of 16 criteria, divided into four main groups: speed, technical and technological, dimensioning, and EEE (economic, exploitation, and environmental) groups.

Considering the modern age that poses challenges and increases uncertainty and the level of risk on a daily basis, decision-making is a very complex and challenging task. Therefore, it is necessary to further develop and apply advanced decision-making models, as is the case in this study, to fulfill research gaps. These gaps mean missing such integrated models which allow for a complete evaluation of each problem in the construction industry. Moreover, modification of any input parameter does not reduce the stability of the model and the possibility of adequate application. An original fuzzy rough model, with a fuzzy model based on D numbers was developed, and an IMF D-SWARA method that uses D numbers to assess the significance of criteria was also developed. Using

D numbers and the algorithm of the IMF SWARA method, it is possible to determine precise coefficients of criteria, taking into account uncertainties and complexities that occur in decision-making processes. The Rough MARCOS method was used to evaluate the performance of construction machinery. The MARCOS method, with its advantages, enables the definition of a precise rank of alternatives with a clear differentiation of their mutual relations. In addition to these advantages, the benefits of Rough Theory were integrated. Thus, contributions can be manifested through the following:

- A novel integrated model for evaluating the performance of construction machinery for sustainable construction was constructed.
- The original multi-criteria model allows for a new extension of the IMF SWARA method that processes fuzzy information based on D numbers and application in the construction industry.
- A flexible multi-criteria model that allows for multiphase verification tests in order to check obtained results related to the evaluation of construction machinery is proposed.

The rest of the paper is structured as follows. In Section 2, we present an extensive review of the literature, with emphasis an on the importance of construction equipment in civil engineering, the application of MCDM methods in the construction industry, and the application of MCDM methods based on D numbers. In Section 3, we showed methodology with a description of the problem and used D numbers, IMF SWARA, and R-MARCOS methods. Section 4 presents the results, along with the detailed calculation steps, of the developed model, while the Section 5 shows validation of the model through sensitivity, comparative analysis, and calculation of correlation tests. The last section summarizes all contributions and shows guidelines for future research.

## 2. Literature Review

### 2.1. Importance of Construction Equipment in Civil Engineering

Wang and Chen [5] presented a method for paver design based on product identity, i.e., based on Kansei engineering and ergonomics. Having in mind the comfort of work and the guarantee of the efficiency, the design of the paver should focus on the physiological and psychological characteristics of people. Thus, it can be concluded that the paver is a leading machine on a construction site, and all other machines that work after it (rollers, trucks with the mixture, etc.) must adapt to its rhythm. Accordingly, Čović [6] pointed out the importance of proper selection and harmonization of the operation of pavers and other machines during the construction of roads in order to achieve the highest quality and economically acceptable construction.

There are many methods of constructing roads and sidewalks, and selecting the best method that will enable faster and more economical construction is a very complex and challenging task. This problem is particularly evident in large projects, as larger projects require accelerated construction processes to be profitable or reduce a negative economic impact caused by traffic jams. Lunkad [7] considered, in more detail, the differences between slip-form pavers and fixed-form pavers, i.e., their advantages and disadvantages. Due to higher production rates, lower labor requirements, and economy, paving using slip-form pavers, compared to conventional fixed-form paving, is widely appreciated as a more efficient way of construction in the modern concrete paving industry. Slip-form pavers differ from conventional fixed-form pavers, as no fixed-forms are required, because slip-form pavers have side forms that move with the machine. Slip-form paving is a process of constructing roads where concrete is extruded by using a paver that forms solid, fresh concrete into a desired slab. Slip-form paver tiling is particularly suitable for time-sensitive projects that require high productivity.

Liu and Wang [8] provided an introduction to key construction technologies and the current foreign development situation. The authors concluded that the construction of pavements by using slip-form pavers is a more efficient mechanical method of construction, compared to the traditional way of constructing/assembling molds for on-site cast because the technology of slip-form structures neglects the site of prefabrication, while

it reduces handling and lifting and can improve the flatness of the structure to obtain better construction results. Moreover, it can achieve "low carbon" construction by reducing energy consumption. It was concluded that the study of new technology for paving the road structure is certainly suitable for improving the structure itself and the quality of the project, increasing labor productivity, reducing construction costs, and significantly extending the life of the structure. In addition to the appropriate selection of the type and model of a paver, it is also important to properly measure the mixture, as indicated by Wang et al. [9]. Mixture proportioning for slip-form paving applications has often been based on recipes or prior mixtures, rather than developing proportions for specific project needs by using local material. Therefore, a performance-based mixture proportioning approach is necessary to balance the target performance requirements for workability, strength, durability, and cost-effectiveness for a given project specification. The aim of this study was to develop an innovative performance-based mixture proportioning method by analyzing the relationship between the characteristics of the selected mixture and their corresponding effects on concrete performance.

In the study performed by Kuntsman et al. [10], a technical and economic comparison of the latest concrete complexes in the construction of hard-type pavements at airfields was conducted. Specifically, various means, i.e., mechanization of domestic and foreign manufacturers used in the construction of hard-type coatings, were studied, and their advantages and disadvantages were considered. The analysis of technical and economic parameters of all analyzed means showed that, for the construction of hard-type coatings at airports in Russia, it is better to use Gomaco sets for concrete paving with electro-hydraulic drives because they have a better concrete paving speed, despite their higher rental cost. Himawati [11] assessed the productivity and operational costs of pavers and dump trucks and compared the calculation and planned results with the actual results obtained during the project. It was concluded that the use of heavy equipment, pavers, and dump trucks is necessary in the process of accelerating work in accordance with the objectives and the time specified, and they discussed how to use heavy equipment efficiently, carefully, and on time. The aforementioned directly affects the productivity of the company in charge of project implementation.

Due to the influence of modern information technology, the construction of traffic infrastructure has entered the "intelligent era". The framework of intelligent construction is presented on all aspects of the integration of modern information technology and conventional highway technologies. The essential elements of intelligent construction include discovery, analysis, decision-making, and execution. A large amount of data, machine learning, and an expert system are applied to provide practical technical solutions for intelligent construction. Chang et al. [12] elaborated the process of intelligent decision-making and automatic control of feedback machines in construction, and finally presented the future development of intelligent construction.

Prokopev et al. [13] studied the concept of cyber–physical road construction system for the continuous non-destructive quality control of asphalt pavement compaction based on artificial intelligence. Information connections between road construction machines on linear structures via Internet protocols enable the optimization of the entire road construction process depending on the environment and the characteristics of asphalt mixture. This helps maximize productivity and reduce construction costs. The study performed by Snyder [14] represents a new modern way of guiding pavers by using "wireless" technology that allows for automatic control of the paver's movement and more precise positioning. This allows contractors to eliminate pins, sensors, and clamp cables, thus significantly saving time and labor required to establish and maintain the system, eliminating safety risks and hazards, reducing the required width of operating space for pavers, and obtaining more precise paving, especially in narrow curves [15].

Bock [16] analyzed the impact of asphalting machines on air pollution, during which the following factors were observed: machine specifications, fuel-consumption data, and load factors of in-use machines. It is very important to evaluate construction machines from

this aspect, as well, because all companies attempt to keep their negative impact as low as possible. The study performed by Ebrahimi et al. [17] aimed to establish a regionalized assessment of the environmental impact of construction machinery equipped with diesel engines in Europe. The obtained results can be useful for decision-making support and for assessing the impact of the transition from fossil fuels to alternative fuels, and the developed methodology provides a basis for future expansion and improvement in this area. Voronov et al. [18] discussed the problem of efficient and high-quality overhaul of machinery and equipment for road construction. The operational characteristics of reliable and precise equipment for road and construction works require high costs to maintain the operability of the machines. Since the replacement of worn-out parts—especially those that are functionally important—is very expensive, in order to ensure the requirements of reliable and safe operation of road construction machines, the authors developed a new mathematical model. Scherbakov et al. [19] studied in more detail the materials and methods used in the manufacturing of construction machinery and their contribution to the mobility and durability of machinery during operation. A similar study was conducted by Abdelmassih et al. [20], where attention was focused on the development of a model that would best describe the processes of deformation of metal parts on construction machinery.

Prochorov [21] believes that the most important thing is to take into account the energy efficiency of construction machinery and efficient management of all available construction equipment to improve the environmental component of construction and construction quality, reduce maintenance costs, and increase energy savings in general. Mobile construction machines are prone to accidents on a dynamic construction site, as the environment on the construction site is constantly changing and continuous monitoring of safety by people is impossible. For this reason, Wang et al. [22] propose an algorithm for monitoring the safety of mobile construction machines in everyday management. The algorithm is validated in the simulation of a real case when the machine enters the warning zone.

### 2.2. Application of MCDM Methods in the Construction Industry

Multi-criteria decision-making methods are widely used in decision-making processes in this area. So far, MCDM has been used to solve many different problems in the construction industry, including all phases of project planning and implementation. Antoniou and Aretoulis [23] performed a comparative analysis of multi-criteria decision-making methods when selecting the types of contracts for the construction of a highway in Greece. To compensate the contractors for the construction of the highway is a complex decision that needs to be made on the basis of scientific evidence. For this reason, seven different types of contracts were analyzed by using the MCDM method based on nine selection criteria. Ighravwe and Oke [24] discussed how to select an adequate public building maintenance strategy by using new multi-criteria decision-making models to confirm their suitability for sustainable practice. The authors used a model that integrates SWARA, WASPAS (Weighted Aggregated Sum Product Assessment), FAD (Fuzzy Axiomatic Design), and ARAS (Additive Ratio Assessment) method to solve this problem. This study draws attention to MCDM methods from the perspectives of the old literature on risks and life-cycle methods in the strategic choice of public building maintenance. Torres-Machi et al. [25] applied the AHP (Analytical Hierarchy Process) method and CBA (Choosing by Advantages) with the aim of assessing the sustainability of road engineering alternatives from the environmental aspect, while Slebi-Acevedo et al. [26] used WASPAS, TOPSIS (Technique for Order of Preference by Similarity to Ideal Solution), and Fuzzy AHP method to select the best fiber to be used in reinforced asphalt mixtures. Akhanova et al. [27] applied MCDM methods for assessing the sustainability of buildings from the environmental aspect in Kazakhstan.

Using the principle of Pareto optimal decisions, Anysz et al. [28] developed a new model of multi-criteria decision-making with cost criteria analysis in the final phase of MCDM. Therefore, during the selection of materials for the construction of walls and the selection of facade systems, it was proposed to exclude the criterion of construction

costs from the analysis and consider it in the final phase of the decision-making process. The authors recommended the application of the previously proposed model because it brings numerous advantages, such as versatility, economic sensitivity, ease of application, and time saving in analysis. Kishore et al. [29] used the AHP and SAW (Simple Additive Weighting) method when forming a framework for contractor selection in construction projects. Overviews of the application of MCDM methods in the construction industry are presented in Table 1.

**Table 1.** Overview of the application of MCDM methods in the construction industry.

| Location of Case Study | Method | Findings | Evaluation Criteria | Authors |
|---|---|---|---|---|
| Taiwan | AHP | Assessment of the level of environmental sustainability of engineering projects for the construction of transport infrastructure. | Performance criteria, environmental criteria, and cost criteria. | Yang et al. [30] |
| India | Fuzzy TOPSIS and BWM | Assessment and selection of sustainable construction materials. | Twenty-three sub-criteria of environmental, economic, and social sustainability. | Mathiyazhagan et al. [31] |
| Spain | SAW, COPRAS, TOPSIS, VIKOR, and MIVES | Sustainability assessment of various modern construction techniques. | A set of 38 indicators related to the economic and environmental characteristics of design and social impact. | Sánchez-Garrido et al. [32] |
| Turkey | AHP | Selection of construction project management models. | Performance, technical experience, financial stability, management performance/qualifications of employees, capacity, safety records, and equipment operation. | Erdogan et al. [33] |
| Iran | Delphi, DEMATEL, ANP, and TOPSIS | Productivity estimation of prefabricated building systems. | Management criteria, planning criteria, and cost criteria. | Shahpari et al. [34] |
| Malaysia | Fuzzy ANP and DEMATEL | Assessment and selection of environmentally friendly building materials. | Criteria for environmental, economic, and social sustainability. | Khoshnava et al. [35] |
| Taiwan | Entropy, AHP, and TOPSIS | Selection of construction material suppliers. | Qualified product rate, product price, product market share rate, supply capacity, new product development rate, delivery time, and delivery time ratio. | Chen [36] |
| Serbia | FUCOM and Fuzzy MABAC | Selection of a location to build a Bailey bridge. | Access roads, scope of work on site arrangement, properties of banks, width of water barrier, masking conditions, scope of works on joining access roads with the crossing point, and protection of units. | Bozanic et al. [37] |
| Spain | WASPAS, TOPSIS, and Fuzzy AHP | Selection of fibers for strengthening reinforced asphalt mixtures. | Volume properties, resistance, strength, service life, stability, sensitivity to moisture, and strength at low temperatures. | Slebi-Acevedo et al. [26] |
| Montenegro | VIKOR and CP | Selection of the optimal combination of groundwork machines. | Practical performance indicators, price of machine combination operating hours, and reliability of machine combinations in relation to age. | Jovanović [38] |
| Iran | CRITIC and EDAS | Prequalification assessment of construction contractors. | Fifty-six criteria related to general information, financial and technical information, information on equipment, management, and professional experience. | Naik et al. [39] |
| Colombia | SD and AHP | Comparison of some strategies employed in the development of sustainable road maintenance policies. | Four criteria: growth of the road network, technical performance, costs, and environmental impact. | Ruiz and Guevara [40] |

Best Worst Method (BWM); the Complex Proportional Assessment (COPRAS); VIšekriterijumsko KOmpromisno Rangiranje (VIKOR); Integrated Value Method for Sustainability Evaluation (MIVES); decision-making trial and evaluation laboratory (DEMATEL); Analytic Network Process (ANP); Full Consistency Method (FUCOM); Multi-Attributive Border Approximation Area Comparison (MABAC); Compromise Programming (CP); Criteria Importance through Intercriteria Correlation (CRITIC); Evaluation Based on Distance from Average Solution (EDAS); System Dynamics (SD); Analytic Hierarchical Process (AHP).

*2.3. The Application of MCDM Methods Based on D Numbers*

Božanić et al. [3] presented a hybrid model for decision-making support based on D numbers, the FUCOM method, and fuzzy RAFSI (Ranking of Alternatives through Functional Mapping of Criterion Subintervals into a Single Interval) method, which is used to solve the selection of a group of construction machines to enable mobility. By applying D numbers, the input parameters for calculating the weight coefficients of the criteria were provided. The calculation of the weight coefficients of the criteria was performed by using the FUCOM method. The best alternative was selected by using fuzzy RAFSI. In the classical Dempster–Schafer theory of evidence, there are limitations attempted to be solved by applying D numbers. Thus, the application of D numbers represents an extension of the Dempster–Schaefer theory of evidence [41,42]. Pribićević et al. [43] used an integrated fuzzy DEMATEL-D model with the aim of forming a multi-criteria framework that would enable objective processing of uncertain linguistic information in a pairwise comparison of criteria. Salimi and Edalatpanah [44] selected suppliers by using the AHP method and D numbers. The proposed methodology has shown high flexibility and a new way in which decision-making based on uncertain information could be improved. Lin et al. [45] analyzed the risks of new energy systems in China by using a model that integrates D numbers and the DEMATEL method. It was concluded that the proposed model, i.e., the application of D numbers, greatly improves existing methods through more precise processing of data and information used in decision-making, and accordingly can find application in solving various problems. Mousavi-Nasab and Sotoudeh-Anvari [46] evaluated renewable energy resources by using the integrated BWM–COPRAS–WASPAS method based on D numbers, while Liu et al. [47] analyzed potential process errors in systems, design, and products, using MCMD-D model numbers.

Efficient supply chain management is essential for any industry in order to achieve the desired level of stability and productivity to meet customer requirements. The selection of the most suitable suppliers is an integral part of the supply chain management that can be efficiently solved by applying various multi-criteria decision-making techniques. For this reason, Chakraborty et al. [42] developed an integrated D-MARCOS model for supplier selection in the iron and steel industry based on seven selective evaluation criteria and the opinions of three decision-makers. By applying D numbers in this model, all ambiguities and uncertainties related to data, which appear during subjective decision-making, were solved. Thus, the use of D numbers increases the ability of decision-makers when handling uncertain information. The MARCOS method was used to rank suppliers based on defined criteria. Zhao and Deng [48] used D numbers to assess the impact of human error on reliability when selecting an optimal contractor. The experimental results showed that the proposed model solves the previously mentioned problem in a very efficient and simple way. In a study performed by Lai and Liao [49], the blockchain platform was evaluated by using the DNMA (Double-Normalization-Based Multiple Aggregation) –CRITIC method based on D numbers. The authors also performed a sensitivity analysis that showed the robustness and stability of the developed method. A comparative analysis showed that the applied method can effectively identify potentially significant criteria in a decision-making process. The introduction of uncertainty theory in the field of decision-making provides a solution for processing qualitative information. Liu et al. [50] believe that the theory of D numbers best solves the previously mentioned problems that arise during the decision-making process, as evidenced by the results of their research in which this approach was applied to analyze and select the best supplier. In the process of handling MCDM problems, due to people's subjective judgment, this inevitably involves various uncertainties, such as inaccuracy, ambiguity, and incompleteness. Xiao [51] believes that D numbers, as a reliable and effective expression of uncertain information, have good performance for handling these types of uncertainties, but that there are still some areas that need to be further explored. Therefore, a new integrated Entropy-D numbers method has been proposed for these problems.

In addition to the previously mentioned areas of application of MCDM methods based on D numbers, so far, they have been also used to solve the following problems: selection of suppliers in the tractor manufacturing industry [52], assessment of "green" supply chain management [53], and selection of automatic cannon for integration into combat vehicles [54] for the assessment of healthcare waste treatment technologies [55].

## 3. Methodology

Roads are divided into categories according to several criteria, such as geopolitical criterion, exploitation criterion, technical criterion, traffic volume, etc. According to the exploitation criterion, roads in Serbia are divided into state roads of the first order, meaning those that form the basis of the road network and represent the connection between Serbia and neighboring countries; and state roads of second order, meaning those connecting regional centers and municipal roads. According to the volume of traffic, roads are divided into highways and class I roads with average annual daily traffic (AADT) greater than 12000, class II roads with AADT between 7000 and 12000, class III roads with AADT between 3000 and 7000, class IV roads with AADT between 1000 and 3000, and class V roads with AADT less than 1000. As roads are defined differently by different criteria, for the needs of more comprehensive research, we adopted three categories for which pavers were evaluated. The categories adopted are primarily based on the width of traffic lanes and the width of asphalting, as follows:

1. Category 1—asphalting width is up to 5 m;
2. Category 2—asphalting width is from 5 to 10 m;
3. Category 3—asphalting width is over 10 m.

This paper presents the evaluation of paver performance for the middle category of roads.

### 3.1. Description of the Problem

This part of the paper presents a list of defined criteria for evaluating the performance of pavers as construction machines and a list of alternatives for making decisions regarding their application for the construction of road infrastructure, the quality of which influences the performance of the economic system, as can be seen in Reference [56]. Based on the literature review and consultations with experts in the field of civil engineering, we defined a set of 16 criteria divided into four main groups: speed, technical and technological, dimensioning, and EEE (economic, exploitation, and environmental). We considered a set of a total of 12 alternatives, which are explained below.

### 3.1.1. Definition of Alternatives

The analysis compares pavers of the following manufacturers: Volvo, Caterpillar, and Vögele. The alternatives are presented and described in Table 2.

**Table 2.** Description of alternatives.

| Alternatives | Capacity of Tank (t) | Asphalting Speed (m/min) | Theoretical Performance (t/h) | Width of Asphalting (m) | Asphalt Installation Thickness (cm) |
|---|---|---|---|---|---|
| A1—Volvo P4820D ABG | 12.5 | 20 | 500 | 6.5 | 30 |
| A2—Volvo P6820D ABG | 13.5 | 20 | 700 | 10 | 20 |
| A3—Volvo P5870c ABG | 12 | 40 | 600 | 8 | 30 |
| A4—Volvo P6870c ABG | 12 | 40 | 700 | 9 | 30 |
| A5—CAT AP555F | 14.5 | 25 | 1168 | 6.5–7.5 | 30 |
| A6—CAT AP500F | 14.5 | 25 | 1168 | 6.5 | 30 |
| A7—Vögele SUPER 1600 | 13 | 25 | 600 | 6.3 | 30 |
| A8—Vögele SUPER 1603 | 13 | 18 | 600 | 6.3 | 30 |
| A9—Vögele SUPER 1600-3 | 13 | 24 | 600 | 7.5 | 30 |
| A10—Vögele SUPER 1603-3 | 13 | 18 | 600 | 7 | 30 |
| A11—Vögele SUPER 1800-3 | 13 | 24 | 700 | 10 | 30 |
| A12—Vögele SUPER 1803-3 | 13 | 18 | 700 | 8 | 30 |

### 3.1.2. Definition of Criteria

The list of defined criteria (Table 3) was, as mentioned, created based on the opinion of experts in the field and References [57–64], and brief explanations of the criteria are given below.

**Table 3.** Definition of criteria.

| Main Criteria | Sub-Criteria | Definition |
|---|---|---|
| Speed criteria | $C_1$—Asphalting speed | Asphalting speed is a criterion that defines the efficiency of the paver in terms of what road length can be asphalted in a given period of time. The asphalting speed is most often expressed in meters of paved road per minute (mpm—meters per minute) or feet per minute (fpm). |
| | $C_2$—Transport speed | Transport speed is the speed at which pavers are transported from one place to another. Paver transport speed is expressed in km/h. The maximum transport speed was used in the analyses. |
| | $C_3$—Conveyor speed | Conveyors are mechanisms that transport asphalt mixtures from tanks in which the asphalt mixtures are located. That is why this criterion is significant. Conveyor speed is expressed in meters per minute (mpm). |
| | $C_4$—Drill speed | The augers take the material being delivered by the conveyors and move it outward across the width of the screed. Drill speed is expressed in revolutions per minute (rpm). |
| Technical and technological group | $C_5$—Tank capacity | Tank capacity is the amount of asphalt mixture that can be found in the paver. Tank capacity is expressed in tons. |
| | $C_6$—Engine power | Engine power is a factor which is a driving force of the paver and affects the movement of the paver. Engine power is expressed in Kw. |
| | $C_7$—Type (wheels/caterpillars) | Based on the way of movement, all pavers can be divided into wheel pavers and caterpillar pavers. |
| | $C_8$—Drill diameter | Drills evenly distribute the material in front of the iron. The function of drills enables homogeneous compaction and asphalting. They can be adjusted to required width by adding drill bits. The larger the diameter of the drill, the more asphalt mixture can be distributed in front of the iron. The diameter of the drill is expressed in millimeters (mm). |
| A group of criteria related to dimensioning | $C_9$—Asphalting width | Asphalting width is the width that the paver asphalts in one pass. This width may be different for the same paver depending on the accessories. Asphalting width is given in meters (m). |
| | $C_{10}$—Asphalt installation thickness | Asphalt is a material consisting of binders and stone material. There are several types of asphalt that differ in the grain size of the stone aggregate used for production. Depending on the types of asphalt, there are minimum and maximum technological thicknesses of asphalt. When evaluating pavers, this criterion is reflected in what the maximum thickness is that can be installed by asphalt pavers. The thickness of the asphalt installation is expressed in centimeters (cm). |
| | $C_{11}$—The dimensions of pavers | The dimensions of pavers are important due to the movement of pavers and possible restrictions on movement in relation to the dimensions. The dimensions of pavers are presented in the form of length, width, and height, and all three dimensions are expressed in meters. |
| | $C_{12}$—The weight of pavers | The weight of pavers is important because it affects the execution of works. Weight can be extremely important if working on poorly bearing soil, where heavier pavers can affect higher soil subsidence, while their weight can help compact the asphalt mixture. |

**Table 3.** *Cont.*

| Main Criteria | Sub-Criteria | Definition |
|---|---|---|
| EEE group of criteria | $C_{13}$—Fuel tank—capacity | Fuel tank capacity is expressed in liters (L). Tank capacity affects the continuity of paving. The higher the capacity of the tank, the less interruptions, and vice versa. |
| | $C_{14}$—Theoretical performance | Theoretical performance is the theoretical amount of asphalt mixture that can be installed. The theoretical performance of pavers is expressed in tons of asphalt mixture per time unit (t/h). |
| | $C_{15}$—Gas emissions | During the construction of roads, certain amounts of gases are emitted in all processes, including asphalting with a paver. According to classification, there are six categories: Euro 1, Euro 2, Euro 3, Euro 4, Euro 5, and Euro 6. Vehicles are categorized based on the emission of certain gases. |
| | $C_{16}$—The purchase price | The purchase price is the material value of a paver, which represents its value depending on its properties. The greater the possibility of applying a paver, the more expensive the paver, and vice versa. |

*3.2. D Numbers*

Dempster–Shafer's evidence (DSE) theory is highly efficient in processing uncertain and indeterminate information and is applicable [65]. However, besides the obvious advantages of DSE theory, there are some limitations, including managing contradictions when the evidence is conflicting. Moreover, one of the limitations is the exclusivity of the elements when distinguishing [66]. Researchers have developed *D* numbers to eliminate these limitations that represent an extension of the DFS theory.

To simplify the presentation of the multi-criteria framework, in the following section, through Definitions 1 and 2, the basic settings of *D* numbers are presented.

**Definition 1.** *Let Q be a finite nonempty set, and a D number is a mapping that* $D : \mathbb{Q} \to [0, 1]$*, with*

$$\sum_{\zeta \subseteq Q} D(\zeta) \leq 1 \quad and \quad D(\varnothing) = 0 \tag{1}$$

*where* $\varnothing$ *is an empty set, and* $\zeta$ *is any subset of Q. If* $\sum_{\zeta \subseteq \mathbb{Q}} D(\zeta) \leq 1$*, the presented information is complete; otherwise, the information is incomplete.*

*For the set* $Q = \{\xi_1, \xi_2, \ldots, \xi_i, \xi_j, \ldots, \xi_n\}$*, where* $\xi_i \in R$ *and* $\xi_i \neq \xi_j$ *(when* $i \neq j$*), then D numbers can be represented as* $D = \{(\xi_1, v_1), (\xi_2, v_2) \ldots (\xi_i, v_i), (\xi_j, v_j) \ldots (\xi_n, v_n)\}$*, where the condition that* $v_i > 0$ *and* $\sum_{i=1}^{n} v_i \leq 1$ *is satisfied.*

**Definition 2.** *Let* $D_1$ *and* $D_2$ *be two D numbers of* $D_1 = \{(\xi_1, v_1), \ldots, (\xi_i, v_i), \ldots, (\xi_n, v_n)\}$ *and* $D_2 = \{(\xi_n, v_n), \ldots, (\xi_i, v_i), \ldots, (\xi_1, v_1)\}$ *. Then we can define a rule for the combination of D numbers* $D = D_1 \times D_2$ *as follows:*

$$\begin{cases} D(\varnothing) = 0 \\ D(\aleph) = 1/(1 - Z_D) \times \sum_{B_1 \cap B_2 = B} D_1(\aleph_1) D_2(\aleph_2), \aleph \neq \varnothing \\ \qquad\qquad with \\ Z_D = \frac{1}{R_1 R_2} \times \sum_{\aleph_1 \cap \aleph_2 = \varnothing} D_1(\aleph_1) D_2(\aleph_2) \\ R_1 = \sum_{\aleph_1 \subseteq \Theta} D_1(\aleph_1) \\ R_2 = \sum_{\aleph_2 \subseteq \Theta} D_2(\aleph_2) \end{cases} \tag{2}$$

The rule for the combination of $D$ numbers as given in Equation (2) allows the fusion of uncertain information presented in $D$ numbers. The integration of $D$ numbers obtained by using Equation (2) is performed by using Equation (3).

$$K(D) = \sum_{i=1}^{n} \xi_i v_i; \xi_i \in R^+, v_i > 0 \tag{3}$$

Suppose that, in a multi-criteria model, there is a set of $m$ alternatives ($B_i$) and $n$ criteria ($C_j$) for evaluation. Moreover, suppose that $e$ experts $E = \{E_1, E_2, \ldots, E_e\}$ present their preferences by applying fuzzy linguistic variables from the set $\Psi = \{\Psi_b, b = 1, 2, \ldots, h\}$. Then we can define detailed steps of the multi-criteria framework as follows.

### 3.3. IMF D-SWARA Algorithm

An extended IMF SWARA method with $D$ numbers, i.e., the IMF D-SWARA algorithm, consists of the following steps:

Step 1: Ranking of criteria according to their importance by expert assessment.

Step 2: In group decision-making, $r$ experts $R = \{R_1, R_2, \ldots, R_n\}$ present their preferences by applying fuzzy linguistic variables from Table 4. Starting from the previously determined rank, the relatively smaller significance of the criterion (criterion $C_j$) was determined in relation $\overline{\kappa_j}$ to the previous one ($C_{j-1}$), and this was repeated for each subsequent criterion [67,68].

**Table 4.** Linguistics and the TFN scale evaluation of criteria.

| Linguistic Variable | Abbreviation | TFN Scale |
|---|---|---|
| Absolutely less significant | ALS | (1,1,1) |
| Dominantly less significant | DLS | (0.5,0.667,1) |
| Much less significant | MLS | (0.4,0.5,0.667) |
| Really less significant | RLS | (0.333,0.4,0.5) |
| Less significant | LS | (0.286,0.333,0.4) |
| Moderately less significant | MDLS | (0.25,0.286,0.333) |
| Weakly less significant | WLS | (0.222,0.25,0.286) |
| Equally significant | ES | (0,0,0) |

Step 3: Transformation of fuzzy $D$ linguistic variables in the $\overline{\partial_j}$ matrix. The evaluation of the $C_j$ ($j = 2, \ldots, n$) criteria under the $C_{j-1}$ ($j = 1, 2, \ldots, n-1$) criteria is represented by the $D$ number $D_{\partial_{ij}} = \left\{ (\xi^1_{\partial_{ij}}, v^1_{\partial_{ij}}), \ldots, (\xi^i_{\partial_{ij}}, v^i_{\partial_{ij}}), \ldots, (\xi^h_{\partial_{ij}}, v^h_{\partial_{ij}}) \right\}$, where $\xi^i_{\partial_{ij}}$ represents the fuzzy linguistic variable from Table 2, and $v^i_{\partial_{ij}}$ represents the probability of choosing the fuzzy linguistic variable. By applying the rules for the combination of $D$ numbers (2) and (3), the final values of fuzzy $D$ numbers are transformed into fuzzy values, $\overline{\partial_j} = \left( \partial^l_j, \partial^m_j, \partial^u_j \right)$. Thus, an aggregated fuzzy $D$ matrix $\partial = \left[ \overline{\partial_j} \right]_{n \times 1}$ was obtained.

$$\overline{\partial_j} = \begin{cases} (1,1,1) & j = 1 \\ \overline{\kappa_j} & j > 1 \end{cases} \tag{4}$$

Step 4: Calculation of the weights $\overline{\ell_j}$ (5):

$$\overline{\ell_j} = \begin{cases} (1,1,1) & j = 1 \\ \dfrac{\overline{\ell_{j-1}}}{\overline{\partial_j}} & j > 1 \end{cases} \tag{5}$$

Step 5: Calculation of the fuzzy weight coefficients (6):

$$\overline{w}_j = \frac{\overline{\ell_j}}{\sum\limits_{j=1}^{m} \overline{\ell_j}} \tag{6}$$

where $w_j$ is the fuzzy relative weight of the criteria $j$, and $m$ denotes the total number of criteria.

### 3.4. Rough MARCOS Method

The R-MARCOS method [69,70] was applied to evaluate the performance of construction machinery, and this algorithm consists of the following:

Step 1: The Rough Decision Matrix $(RN(V))$ is organized as follows:

$$RN(V) = \left[ v_{ij}^L, v_{ij}^U \right]_{m \times n} \tag{7}$$

where $v_{ij}$ denotes values of the initial Rough Matrix, which consists of $m$ alternatives and $n$ criteria.

Step 2: The Extended Rough Matrix $RN(EV)$ is arranged by adding anti-ideal $RN(AID)$ and ideal $RN(ID)$ solutions to the matrix.

$$RN(AID) = \left[ v_{aid}^L, v_{aid}^U \right] = \begin{cases} min_i \left[ v_{ij}^L, v_{ij}^U \right] & if\ j \in B \\ max_i \left[ v_{ij}^L, v_{ij}^U \right] & if\ j \in C \end{cases} \tag{8}$$

$$RN(ID) = \left[ v_{id}^L, v_{id}^U \right] = \begin{cases} max_i \left[ v_{ij}^L, v_{ij}^U \right] & if\ j \in B \\ min_i \left[ v_{ij}^L, v_{ij}^U \right] & if\ j \in C \end{cases} \tag{9}$$

where $AID$ is anti-ideal, while $ID$ is ideal solution. In Equations (8) and (9), $B$ and $C$ indicate beneficial and cost criteria, respectively.

Step 3: The Rough Normalized Matrix $RN(T)$ is obtained by Equations (11) and (12):

$$RN(T) = \left[ t_{ij}^L, t_{ij}^U \right]_{m \times n} \tag{10}$$

$$\left[ t_{ij}^L, t_{ij}^U \right] = \left[ \frac{v_{ij}^L}{v_{id}^U}, \frac{v_{ij}^U}{v_{id}^L} \right] if\ j \in B \tag{11}$$

$$\left[ t_{ij}^L, t_{ij}^U \right] = \left[ \frac{v_{id}^L}{v_{ij}^U}, \frac{v_{id}^U}{v_{ij}^L} \right] if\ j \in C \tag{12}$$

where $v_{ij}^L$ and $v_{ij}^U$ are low and upper values from the initial decision matrix, respectively. Elements $v_{id}^L$ and $v_{id}^U$ represents low and upper of ideal solution.

Step 4: The Rough Weighted Normalized Matrix $RN(E)$ is computed by Equation (13):

$$RN(E) = \left[ e_{ij}^L, e_{ij}^U \right] = \left[ t_{ij}^L \times w_j^L, t_{ij}^U \times w_j^U \right] \tag{13}$$

In this step, it is necessary to multiply the values of criteria weights by values from the normalized matrix.

Step 5: $RN(Z)$ is computed by using Equation (14).

$$RN(Z) = \left[ z_i^L, z_i^U \right] = \sum_{j=1}^{n} \left[ e_{ij}^L, e_{ij}^U \right] \tag{14}$$

where $RN(Z)$ represents the sum of the elements of matrix $E$.

Step 6: Rough utility degrees of alternatives $RN(Y_i^-)$ and $RN(Y_i^+)$ are calculated as follows:

$$RN(Y_i^-) = \left[ y_i^{-L}, y_i^{-U} \right] = \left[ \frac{z_i^L}{z_{aid}^U}, \frac{z_i^U}{z_{aid}^L} \right] \tag{15}$$

$$RN(Y_i^+) = \left[ y_i^{+L}, y_i^{+U} \right] = \left[ \frac{z_i^L}{z_{id}^U}, \frac{z_i^U}{z_{id}^L} \right] \tag{16}$$

where $z_i^L$ and $z_i^U$ are low and upper values from the previous summed matrix, respectively. Elements $z_{id}^L$ and $z_{id}^U$ represent low and upper in respect to the ideal solution.

Step 7: Rough utility degrees ($RN(Y_i^-)$ and $RN(Y_i^+)$) are converted into crisp $Y_i^-$ and $Y_i^+$, using Equations (17) and (18):

$$Y_i^- = \frac{y_i^{-L} + y_i^{-U}}{2} \tag{17}$$

$$Y_i^+ = \frac{y_i^{+L} + y_i^{+U}}{2} \tag{18}$$

Step 8: The utility functions in relation to the anti-ideal $f(Y_i^-)$ and ideal $f(Y_i^+)$ solutions are computed by Equations (20) and (21), respectively.

$$f(Y_i) = \frac{Y_i^+ + Y_i^-}{1 + \frac{1 - f(Y_i^+)}{f(Y_i^+)} + \frac{1 - f(Y_i^-)}{f(Y_i^-)}} \tag{19}$$

where we have the following:

$$f(Y_i^-) = \frac{Y_i^+}{Y_i^+ + Y_i^-} \tag{20}$$

$$f(Y_i^+) = \frac{Y_i^-}{Y_i^+ + Y_i^-} \tag{21}$$

Step 9: The alternatives are sorted from the highest utility function to the lowest utility function.

## 4. Results

### 4.1. Application of IMF D-SWARA Algorithm

Four experts, who were divided into two expert groups, participated in the research.

Expert 1 works as a head of the machine park department in a company engaged in road construction. The company is from Novi Sad. This expert has 20 years of experience. Expert 2 works as a manager of construction, transport, and mechanization in a company from Novi Sad that performs works on the construction of roads and other construction facilities. This expert has over 10 years of experience in this business. Expert 3 is a head of laboratories and mechanization and works as an associate of a large number of companies from Serbia that are primarily engaged in the construction of road infrastructure. This expert has 15 years of experience. Expert 4 is a university professor. This expert has 15 years of experience in the design and construction of pavement structures and road infrastructure and other buildings.

Applying the linguistic scale from Table 4 and $D$ numbers, these two expert groups evaluated the criteria in order to determine their weights, which are shown in Table 5.

**Table 5.** Evaluation by experts through two groups.

| $C_j/C_{j-1}$ for main criteria | |
|---|---|
| $C_1 / C_3$ | $D_1$ = {(ES,0.65),(WLS,0.35)}; $D_2$ = {(ES,0.75),(WLS,0.15),(MDLS,0.1)} |
| $C_4/C_1$ | $D_1$ = {(ES,0.1),(WLS,0.9)}; $D_2$ = {(ES,0.15),(WLS,0.7),(MDLS,0.15)} |
| $C_2 / C_4$ | $D_1$ = {(ES,0.7),(WLS,0.25)}; $D_2$ = {(ES,0.6),(WLS,0.3),(MDLS,0.1)} |

| $C_j/C_{j-1}$ for speed criteria | | $C_j/C_{j-1}$ for TT criteria | |
|---|---|---|---|
| $C_2 / C_1$ | $D_1$ = {(MDLS,0.1),(WLS,0.85)}; $D_2$ = {(ES,0.1),(MDLS,0.15),(WLS,0.75)} | $C_2 / C_3$ | $D_1$ = {(ES,0.85),(WLS,0.15)}; $D_2$ = {(ES,0.75),(WLS,0.15);(LS,0.1)} |
| $C_3 / C_2$ | $D_1$ = {(MLS,0.25),(MDLS,0.75)}; $D_2$ = {(ES,0.05),(MDLS,0.8),(WLS,0.15)} | $C_1 / C_2$ | $D_1$ = {(DLS,0.8),(ALS,0.15)}; $D_2$ = {(RLS,0.1),(DLS,0.8),(ALS,0.1)} |
| $C_4 / C_3$ | $D_1$ = {(RLS,0.15),(MDLS,0.80)}; $D_2$ = {(ES,0.1),(RLS,0.2),(MDLS;0.7)} | $C_4 / C_1$ | $D_1$ = {(WLS,0.65),(MDLS,0.3)}; $D_2$ = {(ES,0.1),(WLS,0.25),(MDLS,0.6)} |

| $C_j/C_{j-1}$ for dimensioning criteria | | $C_j/C_{j-1}$ for the EEE group of criteria | |
|---|---|---|---|
| $C_2 / C_1$ | $D_1$ = {(RLS,0.4),(MDLS,0.6)}; $D_2$ = {(LS,0.15),(RLS,0.35),(MDLS,0.5)} | $C_2 / C_4$ | $D_1$ = {(ES,0.65),(WLS,0.35)}; $D_2$ = {(ES,0.5),(WLS,0.3),(RLS,0.2)} |
| $C_3 / C_2$ | $D_1$ = {(ES,0.35),(WLS,0.6)}; $D_2$ = {(ES,0.3),(WLS,0.55),(MDLS,0.1)} | $C_1 / C_2$ | $D_1$ = {(WLS,0.45),(RLS,0.55)}; $D_2$ = {(WLS,0.45),(MDLS,0.15),(RLS,0.4)} |
| $C_4 / C_3$ | $D_1$ = {(MDLS,0.55),(LS,0.45)}; $D_2$ = {(ES;0.05),(MDLS,0.6),(LS,0.35)} | $C_3 / C_1$ | $D_1$ = {(WLS,0.1),(LS,0.9)}; $D_2$ = {(ES,0.1),(LS;0.8),(RLS,0.1)} |

In the following section, the uncertainty from Table 3 is processed by applying the rules for the combination of *D* numbers. Then, using Equation (2), the expert estimates were fused into a unique fuzzy *D* number (Table 6).

**Table 6.** *Sj* matrix with aggregated fuzzy D numbers.

| $C_j/C_{j-1}$ for main criteria | |
|---|---|
| $C_1 / C_3$ | $D$ = {(ES,0.903),(WLS,0.097)} |
| $C_4 / C_1$ | $D$ = {(ES,0.023),(WLS,0.977)} |
| $C_2 / C_4$ | $D$ = {(ES,0.806),(WLS,0.144)} |

| $C_j/C_{j-1}$ for speed criteria | | $C_j/C_{j-1}$ for TT criteria | |
|---|---|---|---|
| $C_2 / C_1$ | $D$ = {(MDLS,0.022),(WLS,0.928)} | $C_2 / C_3$ | $D$ = {(ES,0.966),(WLS,0.034)} |
| $C_3 / C_2$ | $D$ = {(MDLS,1)} | $C_1 / C_2$ | $D$ = {(DLS,0.916),(ALS,0.021)} |
| $C_4 / C_3$ | $D$ = {(RLS,0.048),(MDLS,0.902} | $C_4 / C_1$ | $D$ = {(WLS,0.428),(MDLS,0.474)} |

| $C_j/C_{j-1}$ for dimensioning criteria | | $C_j/C_{j-1}$ for the EEE group of criteria | |
|---|---|---|---|
| $C_2 / C_1$ | $D$ = {(RLS,0.318),(MDLS,0.682)} | $C_2 / C_4$ | $D$ = {(ES,0.756),(WLS,0.244)} |
| $C_3 / C_2$ | $D$ = {(ES,0.2),(WLS,0.627)} | $C_1 / C_2$ | $D$ = {(WLS,0.479),(RLS,0.521)} |
| $C_4 / C_3$ | $D$ = {(MDLS,0.677),(LS,0.323)} | $C_3 / C_1$ | $D$ = {LS,1)} |

Using Equation (3), the fuzzy *D* numbers are transformed into triangular fuzzy numbers. Table 7 presents an aggregated fuzzy matrix.

**Table 7.** Aggregated fuzzy matrix.

| | **Main** | | **Speed** | | **TT** | | **Dimensioning** | | **EEE** |
|---|---|---|---|---|---|---|---|---|---|
| $C_1-C_3$ | (0.022,0.024,0.028) | $C_2-C_1$ | (0.212,0.238,0.272) | $C_2-C_3$ | (0.008,0.009,0.01) | $C_2-C_1$ | (0.277,0.322,0.386) | $C_2-C_4$ | (0.054,0.061,0.07) |
| $C_4-C_1$ | (0.217,0.244,0.279) | $C_3-C_2$ | (0.25,0.286,0.333) | $C_1-C_2$ | (0.479,0.632,0.937) | $C_3-C_2$ | (0.139,0.157,0.179) | $C_1-C_2$ | (0.28,0.328,0.397) |
| $C_2-C_4$ | (0.032,0.036,0.041) | $C_4-C_3$ | (0.242,0.277,0.325) | $C_4-C_1$ | (0.214,0.243,0.28) | $C_4-C_3$ | (0.262,0.301,0.355) | $C_3-C_1$ | (0.286,0.333,0.4) |

Based on the aggregate matrix shown in Table 7, it is possible to apply the steps of the IMF SWARA method, i.e., Equations (4)–(6), in order to calculate first the weights of the main criteria and then all sub-criteria within the main groups. The calculation for the main criteria is shown in Table 8.

**Table 8.** Calculation of the weights of the main criteria, using the IMF D-SWARA method.

| | $\overline{s_j}$ | | | $\overline{k_j}$ | | | $\overline{q_j}$ | | | $\overline{w_j}$ | | | Crisp Value |
|---|---|---|---|---|---|---|---|---|---|---|---|---|---|
| C3 | | | | 1.000 | 1.000 | 1.000 | 1.000 | 1.000 | 1.000 | 0.281 | 0.284 | 0.289 | 0.284 |
| C1 | 0.022 | 0.024 | 0.028 | 1.022 | 1.024 | 1.028 | 0.973 | 0.976 | 0.979 | 0.273 | 0.277 | 0.283 | 0.278 |
| C4 | 0.217 | 0.244 | 0.279 | 1.217 | 1.244 | 1.279 | 0.761 | 0.785 | 0.804 | 0.214 | 0.223 | 0.232 | 0.223 |
| C2 | 0.032 | 0.036 | 0.041 | 1.032 | 1.036 | 1.041 | 0.731 | 0.757 | 0.779 | 0.205 | 0.215 | 0.225 | 0.215 |
| | | | | SUM | 3.464 | 3.518 | 3.562 | | | | | | |

Based on the results shown in Table 8, it can be seen that the expert groups were quite consistent and that the difference among the main criteria was not significant. The final results of the weights of all criteria are presented below. Table 9 presents the results of the comparison of sub-criteria within each group and the final weights of all criteria obtained by multiplying the values of the main criteria with the sub-criteria within each group.

**Table 9.** Calculation of the weights of all criteria, using the IMF D-SWARA method.

| | I | | | | II | | | | III | | | | IV | | |
|---|---|---|---|---|---|---|---|---|---|---|---|---|---|---|---|
| C11 | 0.331 | 0.342 | 0.355 | C21 | 0.159 | 0.197 | 0.231 | C31 | 0.332 | 0.343 | 0.359 | C41 | 0.205 | 0.223 | 0.240 |
| C12 | 0.260 | 0.276 | 0.293 | C22 | 0.308 | 0.321 | 0.342 | C32 | 0.239 | 0.260 | 0.281 | C42 | 0.286 | 0.296 | 0.308 |
| C13 | 0.195 | 0.215 | 0.234 | C23 | 0.311 | 0.324 | 0.345 | C33 | 0.203 | 0.224 | 0.247 | C43 | 0.146 | 0.167 | 0.187 |
| C14 | 0.147 | 0.168 | 0.189 | C24 | 0.124 | 0.158 | 0.191 | C34 | 0.150 | 0.173 | 0.196 | C44 | 0.306 | 0.314 | 0.325 |
| | I | | | | II | | | | III | | | | IV | | |
| C11 | 0.091 | 0.095 | 0.100 | C21 | 0.033 | 0.042 | 0.052 | C31 | 0.093 | 0.098 | 0.104 | C41 | 0.044 | 0.050 | 0.056 |
| C12 | 0.071 | 0.077 | 0.083 | C22 | 0.063 | 0.069 | 0.077 | C32 | 0.067 | 0.074 | 0.081 | C42 | 0.061 | 0.066 | 0.071 |
| C13 | 0.053 | 0.060 | 0.066 | C23 | 0.064 | 0.070 | 0.078 | C33 | 0.057 | 0.064 | 0.071 | C43 | 0.031 | 0.037 | 0.043 |
| C14 | 0.040 | 0.047 | 0.053 | C24 | 0.025 | 0.034 | 0.043 | C34 | 0.042 | 0.049 | 0.056 | C44 | 0.065 | 0.070 | 0.075 |

Based on the weights obtained by using the IMF D-SWARA method, it can be seen that the ninth criterion, asphalting width, and the first criterion, asphalting speed, are the most significant criteria in evaluating the performance of pavers. The third criterion is transport speed, and the fourth is asphalt installation thickness. The cost criterion, the purchase price, is in fifth position, and the least significant criteria are the drill diameter and gas emission, primarily because the performance of all pavers in terms of exhaust emission is a pretty good solution because there are very slight differences.

### 4.2. Evaluation of Alternatives—Application of Rough MARCOS Algorithm

To determine the performance of pavers for the middle category of roads, first all alternatives were evaluated by the four experts in order to start with the calculation of group decision-making. Table 10 shows the evaluation of the performance of pavers for Expert 1 according to linguistic variables: extremely good (EG), very good (VL), good (G), medium good (MG), medium (M), medium poor (MP), poor (P), very poor (VP), extremely poor (EP), and the assessment by other experts is shown in Appendix A in Tables A1–A3.

By applying the main rules of Rough Number Theory, we obtained the initial decision matrix (Table 11), on the basis of which the R-MARCOS method was applied.

**Table 10.** Example of evaluating the performance of pavers for Expert 1.

|  | *C1* | *C2* | *C3* | *C4* | *C5* | *C6* | *C7* | *C8* | *C9* | *C10* | *C11* | *C12* | *C13* | *C14* | *C15* | *C16* |
|---|---|---|---|---|---|---|---|---|---|---|---|---|---|---|---|---|
| *A*1 | VG | G | MG | MG | MG | G | G | VG | VG | EG | G | G | MG | G | G | G |
| *A*2 | VG | G | G | VG | G | VG | G | VG | EG | VG | G | G | G | VG | G | G |
| *A*3 | EG | VG | VG | EG | G | VG | EG | EG | VG | EG | G | VG | G | VG | VG | VG |
| *A*4 | EG | VG | VG | EG | G | VG | EG | EG | EG | EG | G | VG | G | VG | VG | VG |
| *A*5 | EG | G | VG | VG | G | G | G | VG | VG | EG | G | VG | G | EG | VG | MG |
| *A*6 | EG | G | VG | VG | G | G | G | EG | G | EG | G | VG | G | EG | VG | MG |
| *A*7 | EG | VG | VG | VG | G | G | G | VG | VG | EG | VG | G | G | VG | VG | VG |
| *A*8 | VG | VG | EG | VG | G | G | EG | EG | EG | EG | VG | G | G | EG | VG | VG |
| *A*9 | EG | G | EG | VG | G | G | G | VG | EG | EG | VG | G | G | EG | VG | VG |
| *A*10 | VG | EG | EG | VG | G | G | EG | EG | EG | EG | VG | G | G | EG | VG | VG |
| *A*11 | EG | VG | EG | VG | VG | G | G | VG | EG | EG | VG | G | VG | EG | VG | VG |
| *A*12 | VG | VG | EG | VG | G | G | EG | EG | EG | EG | VG | G | G | VG | VG | MG |

**Table 11.** Initial decision matrix for the R-MARCOS method.

|  | *C1* | | *C2* | | *C3* | | *C4* | | *C5* | | *C6* | | *C7* | | *C8* | |
|---|---|---|---|---|---|---|---|---|---|---|---|---|---|---|---|---|
| *A*1 | 6.00 | 7.52 | 4.50 | 6.46 | 5.27 | 6.25 | 5.65 | 6.90 | 6.27 | 7.25 | 6.25 | 6.75 | 7.13 | 7.88 | 7.56 | 7.94 |
| *A*2 | 6.00 | 7.52 | 4.50 | 6.46 | 5.75 | 7.25 | 7.75 | 8.73 | 7.27 | 8.25 | 8.25 | 8.75 | 7.13 | 7.88 | 7.56 | 7.94 |
| *A*3 | 8.13 | 8.88 | 7.10 | 8.35 | 6.00 | 7.52 | 8.13 | 8.88 | 6.59 | 7.42 | 8.25 | 8.75 | 7.59 | 8.42 | 7.59 | 8.42 |
| *A*4 | 8.13 | 8.88 | 7.10 | 8.35 | 6.00 | 7.52 | 7.75 | 8.73 | 6.75 | 8.25 | 8.25 | 8.75 | 7.59 | 8.42 | 7.59 | 8.42 |
| *A*5 | 6.75 | 8.25 | 5.69 | 6.81 | 7.59 | 8.42 | 7.59 | 8.42 | 7.50 | 8.50 | 6.59 | 7.42 | 7.13 | 7.88 | 8.56 | 8.94 |
| *A*6 | 6.75 | 8.25 | 5.63 | 7.29 | 7.59 | 8.42 | 7.75 | 8.73 | 7.27 | 8.25 | 6.59 | 7.42 | 7.25 | 7.75 | 9.00 | 9.00 |
| *A*7 | 6.75 | 8.25 | 4.65 | 7.77 | 7.75 | 8.73 | 7.75 | 8.73 | 7.27 | 8.25 | 7.27 | 8.25 | 7.13 | 7.88 | 8.25 | 8.75 |
| *A*8 | 5.75 | 7.25 | 7.25 | 8.67 | 8.13 | 8.88 | 7.75 | 8.73 | 7.27 | 8.25 | 7.27 | 8.25 | 7.59 | 8.42 | 8.56 | 8.94 |
| *A*9 | 6.75 | 8.25 | 4.50 | 6.46 | 8.13 | 8.88 | 7.75 | 8.73 | 7.27 | 8.25 | 7.27 | 8.25 | 7.13 | 7.88 | 8.25 | 8.75 |
| *A*10 | 6.75 | 8.25 | 4.50 | 6.46 | 8.13 | 8.88 | 7.75 | 8.73 | 7.27 | 8.25 | 7.27 | 8.25 | 7.13 | 7.88 | 8.25 | 8.75 |
| *A*11 | 6.75 | 8.25 | 4.44 | 6.98 | 8.13 | 8.88 | 7.75 | 8.73 | 7.59 | 8.42 | 7.75 | 8.73 | 7.13 | 7.88 | 8.25 | 8.75 |
| *A*12 | 5.75 | 7.25 | 7.25 | 8.67 | 8.13 | 8.88 | 7.75 | 8.73 | 7.27 | 8.25 | 7.75 | 8.73 | 7.59 | 8.42 | 8.56 | 8.94 |
|  | *C9* | | *C10* | | *C11* | | *C12* | | *C13* | | *C14* | | *C15* | | *C16* | |
| *A*1 | 5.65 | 6.90 | 8.25 | 8.75 | 6.25 | 7.67 | 6.10 | 7.35 | 4.63 | 6.29 | 7.56 | 7.94 | 7.00 | 7.00 | 4.65 | 5.90 |
| *A*2 | 8.25 | 8.75 | 6.25 | 7.67 | 4.71 | 7.13 | 6.10 | 7.35 | 6.75 | 7.73 | 7.56 | 7.94 | 7.06 | 7.44 | 4.33 | 5.75 |
| *A*3 | 6.23 | 8.19 | 8.25 | 8.75 | 4.71 | 7.13 | 6.25 | 7.67 | 6.75 | 7.73 | 8.00 | 8.00 | 6.27 | 7.25 | 4.40 | 6.25 |
| *A*4 | 8.25 | 8.75 | 8.25 | 8.75 | 4.71 | 7.13 | 6.25 | 7.67 | 6.75 | 7.73 | 7.56 | 7.94 | 6.27 | 7.25 | 4.40 | 6.25 |
| *A*5 | 5.65 | 6.90 | 8.25 | 8.75 | 4.71 | 7.13 | 6.25 | 7.67 | 4.25 | 6.54 | 9.00 | 9.00 | 6.59 | 7.42 | 2.40 | 4.25 |
| *A*6 | 5.59 | 6.42 | 8.25 | 8.75 | 4.71 | 7.13 | 6.25 | 7.67 | 5.75 | 6.73 | 9.00 | 9.00 | 4.71 | 6.38 | 2.40 | 4.25 |
| *A*7 | 5.65 | 6.90 | 8.25 | 8.75 | 6.25 | 7.67 | 6.65 | 7.90 | 6.10 | 7.35 | 7.13 | 7.88 | 4.71 | 6.38 | 4.71 | 6.38 |
| *A*8 | 5.71 | 7.38 | 8.25 | 8.75 | 5.75 | 7.61 | 6.65 | 7.90 | 6.10 | 7.35 | 7.10 | 8.35 | 4.71 | 6.38 | 4.40 | 6.25 |
| *A*9 | 6.65 | 7.90 | 8.25 | 8.75 | 6.25 | 7.67 | 6.17 | 7.84 | 6.10 | 7.35 | 7.10 | 8.35 | 4.71 | 6.38 | 4.40 | 6.25 |
| *A*10 | 6.65 | 7.90 | 8.25 | 8.75 | 6.25 | 7.67 | 6.17 | 7.84 | 6.10 | 7.35 | 7.10 | 8.35 | 4.71 | 6.38 | 4.40 | 6.25 |
| *A*11 | 8.25 | 8.75 | 8.25 | 8.75 | 5.50 | 7.46 | 6.50 | 8.46 | 8.25 | 8.75 | 7.59 | 8.42 | 4.71 | 6.38 | 4.25 | 5.75 |
| *A*12 | 7.27 | 8.25 | 8.25 | 8.75 | 5.50 | 7.46 | 6.23 | 8.19 | 6.23 | 8.19 | 7.56 | 7.94 | 4.71 | 6.38 | 3.65 | 4.90 |

Based on the application of all steps of the R-MARCOS method, which first includes defining an ideal and anti-ideal solution, normalization of the decision matrix, its weighting with the weights obtained by the IMF D-SWARA method, and defining levels and utility functions in relation to the ideal and anti-ideal solution were performed and are presented in Table 12. It is important to note that all criteria were modeled as benefits.

**Table 12.** The results of developed IMF D-SWARA—Rough MARCOS model.

| | $RN(Z)$ | | $RN(Y_i^-)$ | | $RN(Y_i^-)$ | | $Y_i^-$ | $Y_i^-$ | $f(Y_i^-)$ | $f(Y_i^+)$ | $f(Y_i)$ | **Rank** |
|---|---|---|---|---|---|---|---|---|---|---|---|---|
| *AID* | 0.65 | 0.81 | | | | | | | | | | |
| *A1* | 0.73 | 0.85 | 0.899 | 1.31 | 0.725 | 0.946 | 1.105 | 0.836 | 0.431 | 0.569 | 0.631 | 12 |
| *A2* | 0.77 | 0.91 | 0.952 | 1.39 | 0.767 | 1.003 | 1.171 | 0.885 | 0.43 | 0.57 | 0.668 | 9 |
| *A3* | 0.81 | 0.95 | 1.007 | 1.462 | 0.812 | 1.055 | 1.235 | 0.934 | 0.431 | 0.569 | 0.705 | 3 |
| *A4* | 0.83 | 0.96 | 1.03 | 1.475 | 0.83 | 1.064 | 1.253 | 0.947 | 0.43 | 0.57 | 0.714 | 1 |
| *A5* | 0.75 | 0.89 | 0.928 | 1.37 | 0.749 | 0.989 | 1.149 | 0.869 | 0.431 | 0.569 | 0.656 | 10 |
| *A6* | 0.75 | 0.89 | 0.932 | 1.361 | 0.751 | 0.982 | 1.147 | 0.867 | 0.43 | 0.57 | 0.654 | 11 |
| *A7* | 0.78 | 0.93 | 0.962 | 1.429 | 0.775 | 1.031 | 1.196 | 0.903 | 0.43 | 0.57 | 0.682 | 8 |
| *A8* | 0.79 | 0.94 | 0.979 | 1.442 | 0.789 | 1.041 | 1.211 | 0.915 | 0.43 | 0.57 | 0.69 | 5 |
| *A9* | 0.78 | 0.93 | 0.969 | 1.432 | 0.781 | 1.033 | 1.201 | 0.907 | 0.43 | 0.57 | 0.684 | 6 |
| *A10* | 0.78 | 0.93 | 0.969 | 1.432 | 0.781 | 1.033 | 1.201 | 0.907 | 0.43 | 0.57 | 0.684 | 6 |
| *A11* | 0.82 | 0.96 | 1.011 | 1.469 | 0.815 | 1.06 | 1.24 | 0.938 | 0.431 | 0.569 | 0.708 | 2 |
| *A12* | 0.80 | 0.94 | 0.994 | 1.445 | 0.801 | 1.043 | 1.22 | 0.922 | 0.43 | 0.57 | 0.695 | 4 |
| *İD* | 0.90 | 1.00 | | | | | | | | | | |

The calculated values of paver performance indicate that there is a set of three pavers which, under the given conditions of consideration and decision makers' preferences, can almost equally participate in the construction of road infrastructure, namely: *A4*—Volvo P6870c ABG, *A11*—Vögele SUPER 1800-3, and *A3*—Volvo P6870c ABG. The worst solutions are the first and sixth alternatives. Considering that, based on the final values of the performance of pavers, it can be concluded that there are no drastic differences, and a logical consequence is to perform a sensitivity analysis in order to verify the results.

## 5. Sensitivity, Comparative Analysis, and Discussion

Within the sensitivity analysis and comparative analysis, a multi-phase testing of previously obtained results was performed. In the first phase, the influence of changing the significance of criteria on the ranking of paver performance was examined. The second phase involved the application of a reverse rank wherein the worst alternative was eliminated from the model and repeated to a matrix size of one. The third phase involved a comparative analysis with seven other Rough MCDM methods, while the fourth phase involved the calculation of SCC and WS statistical correlation coefficients for both rankings when changing criterion weights, as well as for rankings in comparative analysis.

### 5.1. Simulation of New Criterion Weights

In this section, we look at 30 scenarios that we created in which the weights of the three most significant criteria, *C9*, *C1*, and *C2*, were changed by applying Equation (22) [71–73]. We decided to create 30 scenarios because these three criteria have the largest influence on the decision-making process, with the assumption that other criteria have not influenced alternative ranking.

$$W_{n\beta} = (1 - W_{n\alpha}) \frac{W_\beta}{(1 - W_n)} \quad (22)$$

The three most significant criteria were reduced by 5% in the 1st, 11th, and 21st scenarios for *C9*, *C1*, and *C2*, respectively, while this value was further reduced by 10% in all other scenarios. Therefore, the range of the decrease in the value of the three most significant criteria is 5–95%. Figure 1 shows all criterion values in 30 scenarios.

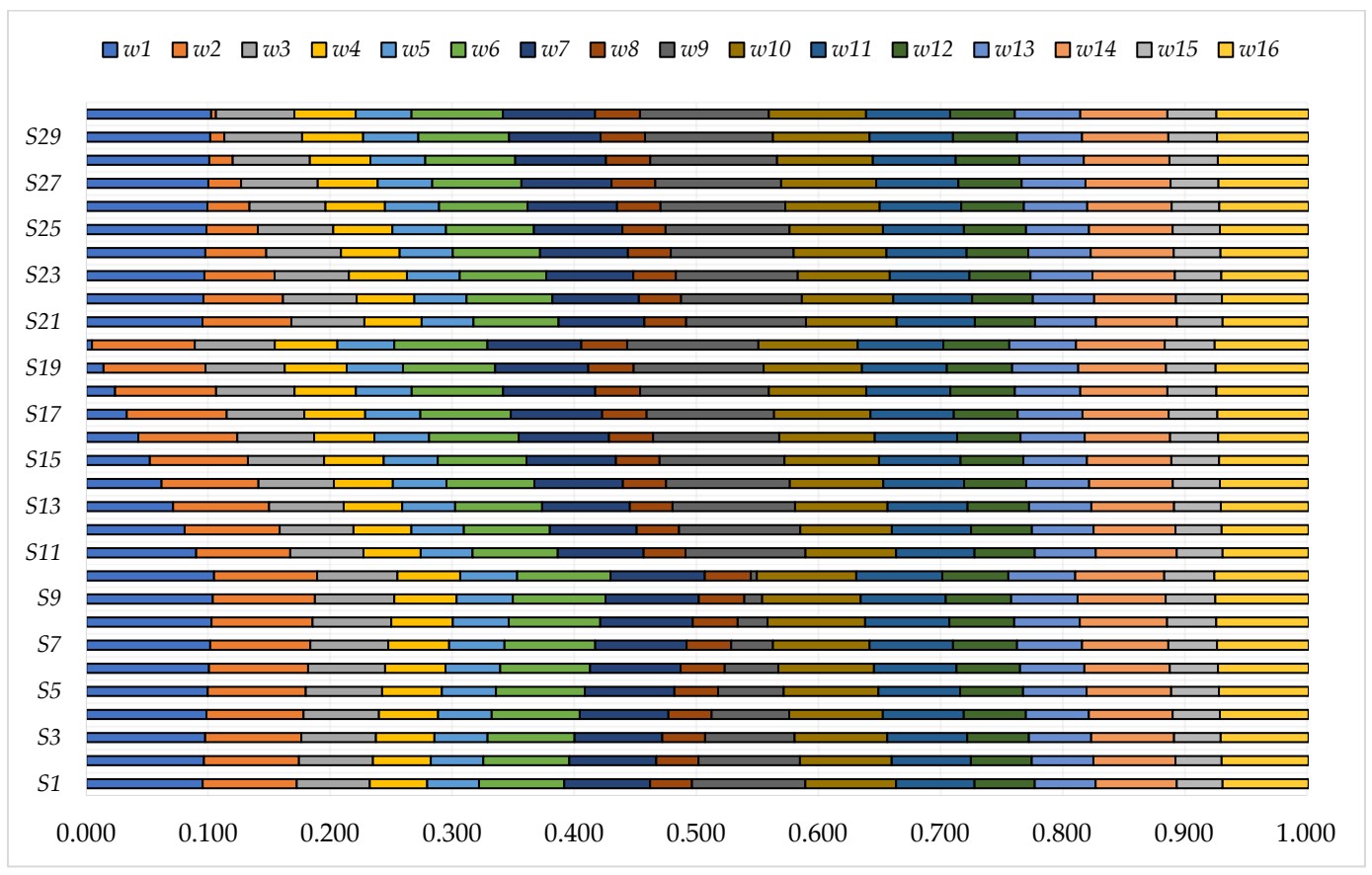

**Figure 1.** New simulated criterion weights across 30 scenarios.

Figure 1 shows the weights of the criteria in new scenarios, e.g., the most significant criterion, *C9*, with its original weight of 0.098 in Scenarios 1–10 (*S1*–*S10*), has the following values 0.093, 0.083, 0.073, 0.064, 0.054, 0.044, 0.034, 0.024, 0.015, and 0.005, respectively. Certainly, as the value of one criterion decreases, the values of all other criteria increase proportionally.

For each scenario, the calculation was performed again by using the R-MARCOS method, so the obtained results are presented in Figure 2.

When observing the obtained results and rankings of paver performance through sensitivity analysis, it is clear that the model is sensitive to changes in the significance of criteria, and this is understandable; however, it is necessary to determine to what extent this is so. For this reason, the SCC [74] and WS [75] correlation coefficients presented in Figure 3 were calculated. In the sensitivity analysis, only the worst alternative did not change its rank, while the others changed their position depending on the scenario. In the first 10 scenarios, when the value of criterion *C9* was reduced, there were no changes in terms of the best *A4* alternative, but in *S10*, *A3* shared the first position. The conclusion is that, with a decrease in the importance of the criterion of asphalting width, and with the increase of the weight of other criteria, the significance of the third, seventh, eighth, and eleventh alternatives increases. By reducing the value of the criterion of asphalting speed in S11–S20, there are no changes when it comes to the two best alternatives, so this criterion does not affect the position of the best alternatives; meanwhile, the changes of other alternatives are by one position. *S21*–*S30* change the ranking of the best alternatives because two best alternatives change their places with drastic reduction of the significance of transport speed criterion in *S27*–*S30*.

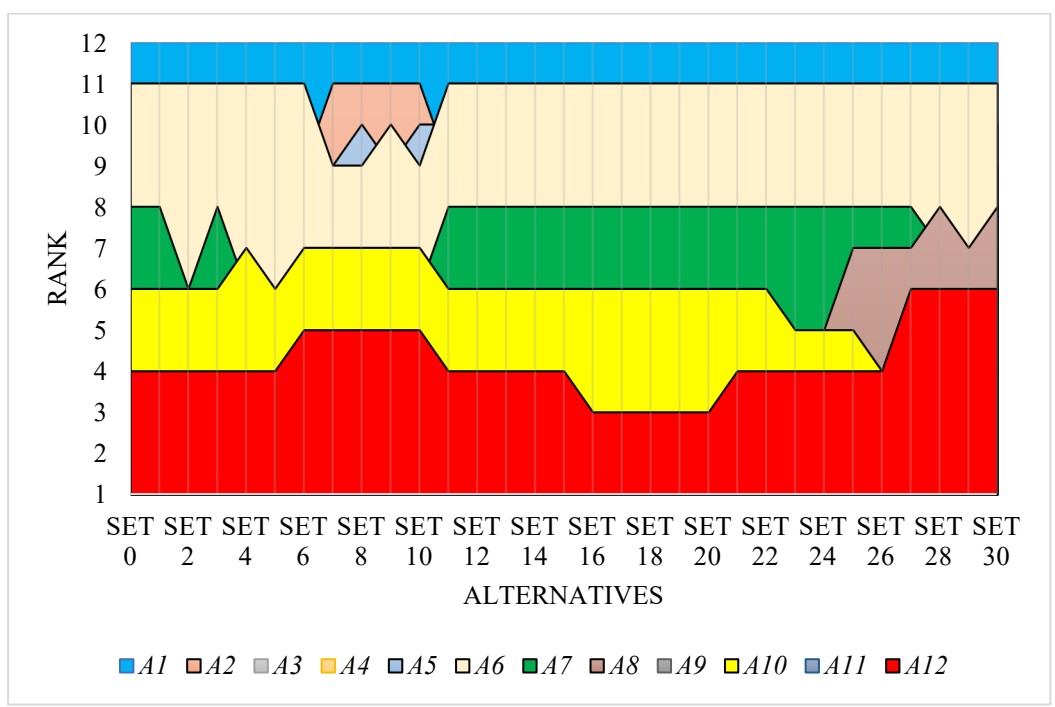

**Figure 2.** Results of the R-MARCOS method for simulated criterion weights.

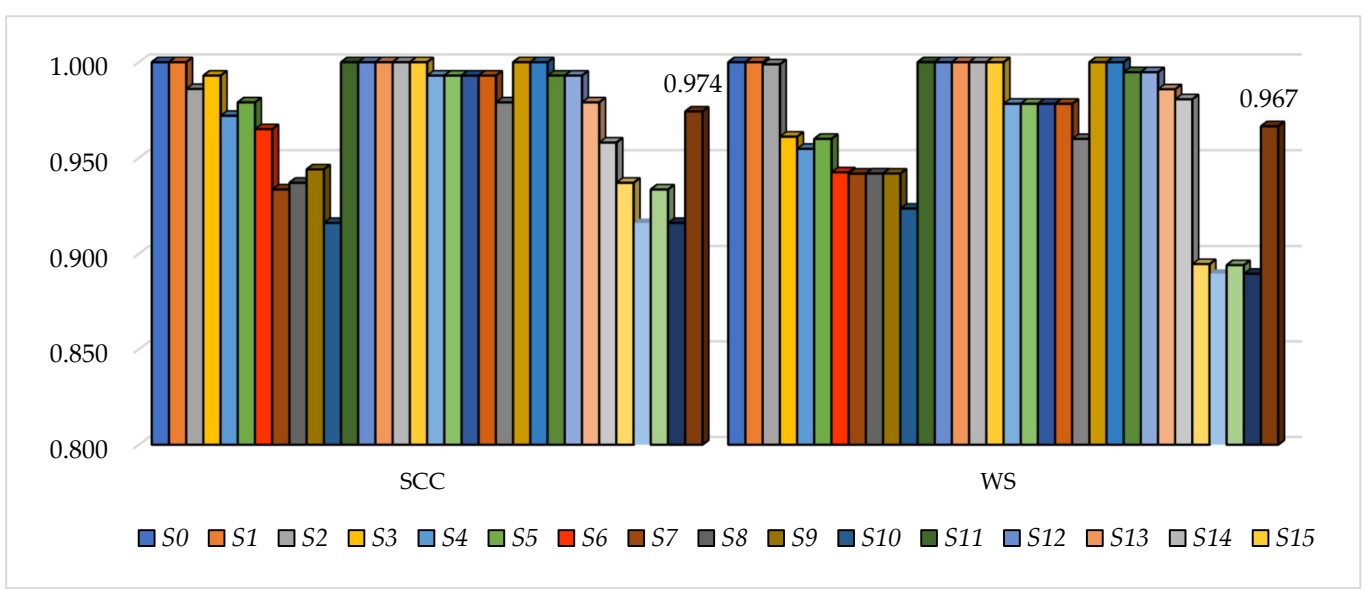

**Figure 3.** SCC and WS coefficients in sensitivity analysis.

*5.2. SCC and WS—Statistical Correlation Coefficients When Changing the Weights of Criteria*

The calculated SCC and WS correlation coefficients for all 30 scenarios in which the weights of the criteria change are presented in Figure 3.

Although there are certain changes in the rankings of alternatives, as previously explained in detail, the calculated correlation coefficients show a high level of rank correlation. The average SCC for all 30 scenarios is 0.974, while the WS is 0.967, which indicates an almost-full correlation. The lowest correlation coefficient for SCC is in S10, S28, and S30 when it is 0.916, while for the WS coefficient, the lowest correlation is in S28 and S30, and it is 0.890, which results in exchanging the positions of two best-ranked alternatives. Given the full correlation, it can be concluded that the model is sensitive to changes in the weights

of criteria, but that the set of three best pavers maintain their good performance, and that the changes in rankings are not of high intensity.

### 5.3. Changing the Size of the Initial Decision Matrix

In this part of performing the validity tests of the results obtained, the size of the initial Rough Decision Matrix was changed. Out of the initial set of pavers, which consists of 12 pavers, the worst alternative is eliminated, and it is Alternative 1 (*A1*) in the first scenario, *A*6 in the second, *A*5 in the third, etc. The size of the initial Rough Matrix is reduced until the best alternative remains the only option, because, in each scenario, one alternative is eliminated which is in the last position. Figure 4 shows the results of changing the size of the decision matrix from two aspects: the first refers to changes in the values of pavers, while the second refers to the rankings in the scenarios created.

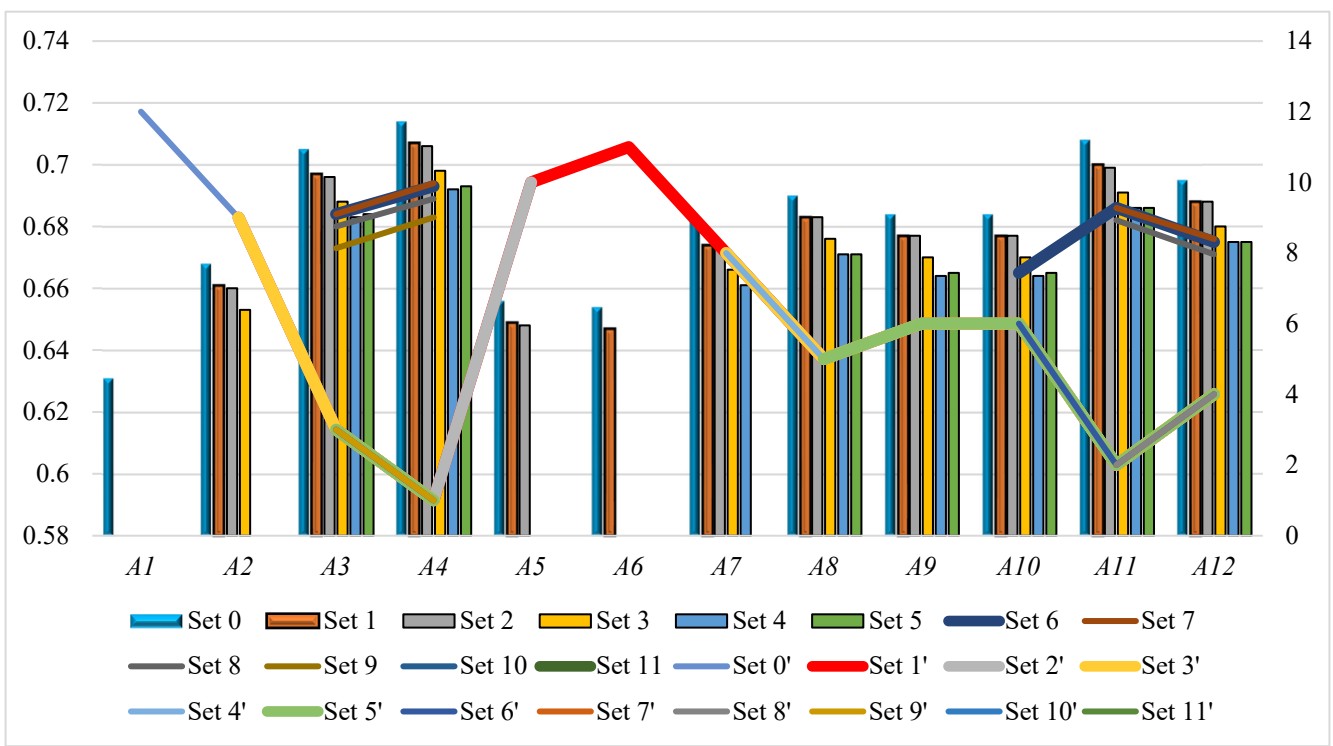

**Figure 4.** Validity test results with changing the size of the initial Rough Decision Matrix.

According to the testing and the results shown in Figure 4, the complete stability of the results obtained can be confirmed because there is no change in ranking in any scenario, which is proof of the validity of the developed IMF D-SWARA-R-MARCOS model. Even the paver performance values do not change too much, but they are in a range of about 10%.

### 5.4. Additional Comparative Analysis with Rough MCDM Methods

In the fourth part of testing the validity of the proposed model, a comparative analysis was performed, along with seven other Rough MCDM methods: R-MABAC [76], R-TOPSIS [77], R-WASPAS [78], R-ARAS [79], R-SAW [80], R-COPRAS [81], and R-CoCoSo [82]. Figure 5 shows the results of the comparative analysis, including the performance values for each method and the final rank.

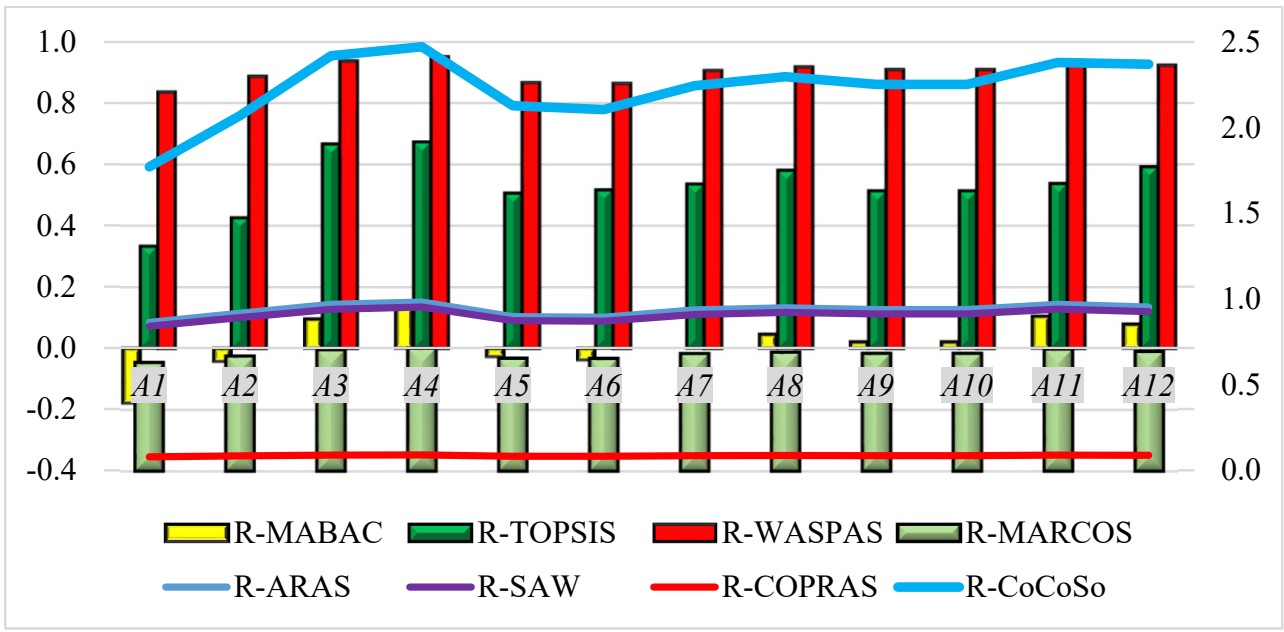

**Figure 5.** Validity test results in the part of comparative analysis.

In the comparative analysis, depending on the application of a method, the ranks of pavers change, thus further causing the calculation of the SCC and WS coefficients for determining the correlation in this part of the verification of previously obtained results. It is important to note that A4 retains the best position in each method, as well as A1, which is always in the worst position. The greatest change happens by applying the R-TOPSIS method when the sixth alternative changes its position by four places, advancing from 11th to 7th place. Other changes in the rankings of alternatives do not imply large oscillations, and this is partly explained below through statistical correlation.

### 5.5. SCC and WS—Statistical Correlation Coefficients in Comparative Analysis

Table 13 shows the correlation values for the overall comparative analysis, together with the average values for both SCC and WS coefficients.

**Table 13.** SCC and WS coefficients for rank changes in comparative analysis.

| SCC | R-MARCOS | R-MABAC | R-TOPSIS | R-WASPAS | R-ARAS | R-SAW | R-COPRAS | R-CoCoSo | AV |
|---|---|---|---|---|---|---|---|---|---|
| R-MARCOS | 1.000 | 0.979 | 0.846 | 1.000 | 0.993 | 1.000 | 0.993 | 0.972 | 0.973 |
| R-MABAC | 0.979 | 1.000 | 0.881 | 0.979 | 0.972 | 0.979 | 0.972 | 0.993 | 0.969 |
| R-TOPSIS | 0.846 | 0.881 | 1.000 | 0.846 | 0.867 | 0.846 | 0.867 | 0.902 | 0.882 |
| R-WASPAS | 1.000 | 0.979 | 0.846 | 1.000 | 0.993 | 1.000 | 0.993 | 0.972 | 0.973 |
| R-ARAS | 0.993 | 0.972 | 0.867 | 0.993 | 1.000 | 0.993 | 1.000 | 0.979 | 0.975 |
| R-SAW | 1.000 | 0.979 | 0.846 | 1.000 | 0.993 | 1.000 | 0.993 | 0.972 | 0.973 |
| R-COPRAS | 0.993 | 0.972 | 0.867 | 0.993 | 1.000 | 0.993 | 1.000 | 0.979 | 0.975 |
| R-CoCoSo | 0.972 | 0.993 | 0.902 | 0.972 | 0.979 | 0.972 | 0.979 | 1.000 | 0.971 |
| | | | | | | | | | 0.961 |
| **WS** | **R-MARCOS** | **R-MABAC** | **R-TOPSIS** | **R-WASPAS** | **R-ARAS** | **R-SAW** | **R-COPRAS** | **R-CoCoSo** | **AV** |
| R-MARCOS | 1.000 | 0.999 | 0.887 | 1.000 | 0.961 | 1.000 | 0.961 | 0.960 | 0.971 |
| R-MABAC | 1.000 | 1.000 | 0.887 | 1.000 | 0.961 | 1.000 | 0.961 | 0.961 | 0.971 |
| R-TOPSIS | 0.927 | 0.928 | 1.000 | 0.927 | 0.957 | 0.927 | 0.957 | 0.958 | 0.948 |
| R-WASPAS | 1.000 | 0.999 | 0.887 | 1.000 | 0.961 | 1.000 | 0.961 | 0.960 | 0.971 |
| R-ARAS | 0.961 | 0.960 | 0.948 | 0.961 | 1.000 | 0.961 | 1.000 | 0.999 | 0.974 |
| R-SAW | 1.000 | 0.999 | 0.887 | 1.000 | 0.961 | 1.000 | 0.961 | 0.960 | 0.971 |
| R-COPRAS | 0.961 | 0.960 | 0.948 | 0.961 | 1.000 | 0.961 | 1.000 | 0.999 | 0.974 |
| R-CoCoSo | 0.961 | 0.961 | 0.948 | 0.961 | 1.000 | 0.961 | 1.000 | 1.000 | 0.974 |
| | | | | | | | | | 0.969 |

The calculated correlation coefficients show a very high correlation of ranks when applying seven different methods. The SCC coefficient is 0.961, while the WS is 0.969,

which are extremely high correlation coefficients, especially if the size of the comparative analysis is taken into account. The proposed IMF D-SWARA-R-MARCOS model has a full correlation with IMF D-SWARA-R-WASPAS and IMF D-SWARA-SAW. With other methods, except for R-TOPSIS (SCC = 0.864, WS = 0.887), it has an extremely high correlation coefficient, which confirms the stability of the results obtained.

## 6. Conclusions

Through the research we conducted and them presented in this paper, we developed an intelligent integrated MCDM model for evaluating the performance of construction machinery as one of the most important elements in achieving the high quality of road infrastructure construction. High-quality decision-making and selection of adequate mechanization reduces the total costs in the field of civil engineering, and there is an obvious need to conduct such an analysis. The case study was performed for the middle category of roads in the territory of Serbia, based on expert preferences, that is part of a larger and more complex study. In addition to the possibility of reducing costs by adequate selection of pavers in a set of complex and conflicting criteria, which is the goal of the study, the greatest contribution is reflected in a novel integrated Fuzzy *D* numbers–Rough MCDM model. This model enables it users to overcome uncertainties and inaccuracies in evaluating the performance of construction machinery and is completely flexible for changing any parameter and application in other areas. We considered a set of 12 pavers, which were evaluated on the basis of 16 criteria divided into four equal structures. The obtained results based on the developed IMF D-SWARA-R-MARCOS model show that three pavers can participate almost equally in the construction of road infrastructure, namely *A4*—Volvo P6870c ABG, *A11*—Vögele SUPER 1800-3, and *A3*—Volvo P6870c ABG. The developed IMF D-SWARA-R-MARCOS model was verified through a multi-phase validity test, which includes changing the impact of the most significant criteria, reducing the size of the initial matrix, conducting a comparative analysis with other methods, and determining statistical correlation.

Future research certainly implies the continuation of the application of this model for roads of a lower category, as well as for highways, as this would expand this research. Moreover, as part of future research, there is a need for high-quality decision-making for other construction machinery and equipment or the formation of a set of machines, and their selection.

**Author Contributions:** Conceptualization, B.M. and M.M.; methodology, Ž.S. and B.M.; validation, S.J. and S.S.; formal analysis, S.J. and M.M.; investigation, M.M. and S.S.; writing—original draft preparation, Ž.S. and B.M.; writing—review and editing, S.J. All authors have read and agreed to the published version of the manuscript.

**Funding:** This research received no external funding.

**Institutional Review Board Statement:** Not applicable.

**Informed Consent Statement:** Not applicable.

**Data Availability Statement:** Not applicable.

**Conflicts of Interest:** The authors declare no conflict of interest.

## Appendix A

**Table A1.** Evaluating the performance of pavers for Expert 2.

|     | C1 | C2 | C3 | C4 | C5 | C6 | C7 | C8 | C9 | C10 | C11 | C12 | C13 | C14 | C15 | C16 |
|-----|----|----|----|----|----|----|----|----|----|-----|-----|-----|-----|-----|-----|-----|
| A1  | VG | G  | G  | VG | VG | MG | EG | VG | M  | EG  | VG  | VG  | P   | VG  | G   | M   |
| A2  | VG | G  | VG | EG | EG | EG | EG | VG | EG | VG  | VG  | G   | MG  | VG  | G   | M   |
| A3  | EG | VG | VG | EG | VG | EG | G  | VG | EG | EG  | VG  | G   | MG  | VG  | MG  | M   |
| A4  | EG | VG | VG | EG | VG | EG | G  | VG | EG | EG  | VG  | G   | MG  | VG  | MG  | M   |
| A5  | VG | EG | EG | EG | EG | VG | EG | EG | M  | EG  | VG  | G   | VP  | EG  | MG  | P   |
| A6  | VG | G  | EG | EG | EG | VG | G  | EG | M  | EG  | VG  | G   | M   | EG  | M   | P   |
| A7  | VG | EG | EG | EG | EG | EG | EG | EG | M  | EG  | G   | MG  | M   | VG  | M   | M   |
| A8  | G  | EG | EG | EG | EG | EG | G  | EG | M  | EG  | VG  | MG  | M   | VG  | M   | M   |
| A9  | VG | G  | EG | EG | EG | EG | EG | EG | G  | EG  | G   | M   | M   | VG  | M   | M   |
| A10 | G  | EG | EG | EG | EG | EG | G  | EG | G  | EG  | VG  | MG  | MG  | VG  | M   | M   |
| A11 | VG | G  | EG | EG | EG | EG | EG | EG | EG | EG  | VG  | M   | VG  | VG  | M   | MP  |
| A12 | G  | EG | EG | EG | EG | EG | G  | EG | VG | EG  | VG  | M   | M   | VG  | M   | MP  |

**Table A2.** Evaluating the performance of pavers for Expert 3.

|     | C1 | C2 | C3 | C4 | C5 | C6 | C7 | C8 | C9 | C10 | C11 | C12 | C13 | C14 | C15 | C16 |
|-----|----|----|----|----|----|----|----|----|----|-----|-----|-----|-----|-----|-----|-----|
| A1  | M  | P  | M  | MG | MG | G  | G  | VG | MG | VG  | VG  | G   | G   | VG  | G   | MP  |
| A2  | M  | P  | M  | G  | G  | VG | G  | VG | VG | M   | MG  | VG  | VG  | VG  | G   | MP  |
| A3  | G  | MG | M  | G  | MG | VG | VG | VG | G  | VG  | MG  | VG  | VG  | VG  | G   | MP  |
| A4  | G  | MG | M  | G  | MG | VG | VG | VG | VG | VG  | MG  | VG  | VG  | VG  | G   | MP  |
| A5  | MG | MP | G  | G  | G  | G  | G  | EG | MG | VG  | MG  | VG  | G   | EG  | G   | VP  |
| A6  | MG | MP | G  | G  | G  | G  | VG | EG | MG | VG  | MG  | VG  | G   | EG  | M   | VP  |
| A7  | MG | P  | G  | G  | G  | VG | G  | EG | MG | VG  | VG  | EG  | VG  | VG  | M   | MP  |
| A8  | M  | MG | G  | G  | G  | VG | VG | EG | MG | VG  | G   | EG  | VG  | VG  | M   | MP  |
| A9  | MG | P  | G  | G  | G  | VG | G  | EG | MG | VG  | VG  | EG  | VG  | VG  | M   | MP  |
| A10 | M  | MG | G  | G  | G  | VG | VG | EG | MG | VG  | G   | EG  | VG  | VG  | M   | MP  |
| A11 | MG | P  | G  | G  | G  | VG | G  | EG | VG | VG  | MG  | EG  | EG  | VG  | M   | MP  |
| A12 | M  | MG | G  | G  | G  | VG | VG | EG | G  | VG  | MG  | EG  | VG  | VG  | M   | MP  |

**Table A3.** Evaluating the performance of pavers for Expert 4.

|     | C1 | C2 | C3 | C4 | C5 | C6 | C7 | C8 | C9 | C10 | C11 | C12 | C13 | C14 | C15 | C16 |
|-----|----|----|----|----|----|----|----|----|----|-----|-----|-----|-----|-----|-----|-----|
| A1  | MG | M  | M  | M  | G  | MG | G  | G  | MG | VG  | M   | M   | MG  | MG  | G   | M   |
| A2  | MG | M  | MG | EG | VG | EG | G  | G  | VG | G   | P   | M   | VG  | G   | VG  | MP  |
| A3  | EG | EG | MG | EG | G  | EG | VG | G  | G  | VG  | P   | M   | VG  | MG  | MG  | MP  |
| A4  | EG | EG | MG | VG | EG | EG | VG | G  | VG | VG  | P   | M   | VG  | G   | MG  | MP  |
| A5  | G  | G  | VG | VG | EG | MG | G  | EG | MG | VG  | P   | M   | MG  | EG  | G   | VP  |
| A6  | G  | VG | VG | EG | VG | MG | VG | EG | MG | VG  | P   | M   | MG  | EG  | MP  | VP  |
| A7  | G  | M  | EG | EG | VG | G  | G  | VG | MG | VG  | M   | G   | G   | MG  | MP  | M   |
| A8  | MG | EG | EG | EG | VG | G  | VG | VG | MG | VG  | MP  | G   | G   | MG  | MP  | MP  |
| A9  | G  | M  | EG | EG | VG | G  | G  | VG | G  | VG  | M   | G   | G   | MG  | MP  | MP  |
| A10 | MG | EG | EG | EG | VG | G  | VG | VG | G  | VG  | MP  | G   | G   | MG  | MP  | MP  |
| A11 | G  | M  | EG | EG | VG | EG | G  | VG | VG | VG  | MP  | EG  | EG  | G   | MP  | MP  |
| A12 | MG | EG | EG | EG | VG | EG | VG | VG | G  | VG  | MP  | VG  | VG  | G   | MP  | P   |

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
