# Peer review of "Intelligent Novel IMF D-SWARA—Rough MARCOS Algorithm for Selection Construction Machinery for Sustainable Construction of Road Infrastructure"

_buildings, doi:10.3390/buildings12071059_

Round 1

Reviewer 1 Report

- The paper "Intelligent Novel IMF D-SWARA – Rough MARCOS Algorithm for Improvement of Performance for Construction of Road Infrastructure" is well written manuscript with a newly proposed integrated MCDM methodology.

- The authors are very familiar with the topics covered in the paper. It is well written with all the necessary elements. It is easy to read because is synthesized in a good manner.

Some of the advantages of the paper are:

- Clearly abstract with all core elements.

- The overall quality of the manuscript.

- The authors have ensured well paper structure with a clear explanation. In the introduction, they show the significance of the field first, and after that clearly aims and contributions. A literature review is very extensive and it is combined text with a tabular review which increases the quality of this section. MCDM problem is well defined with an explanation of basic elements. The methodology is novel and appropriate for solving any MCDM problem. Extensive sensitivity and comparative analysis is one more proof of excellent work.

- Developing of novel methodology with clear contributions.

- Well description of variants and used criteria.

The paper has great potential and can be accepted after the following minor corrections:

- Add the structure of the paper as the last paragraph in the introduction.

-Some similar papers should be discussed in the manuscript as follows: (i) An analytics approach to decision alternative prioritization for zero-emission zone logistics (ii) Prioritizing Transport Planning Strategies for Freight Companies Towards Zero Carbon Emission Using Ordinal Priority Approach

- In Table 8 you show an example of evaluating the performance of pavers for Expert 1. It will be useful if you add in appendix evaluation by the rest experts (E2, E3, and E4).

- Why did you create 30 scenarios in validation analysis? Please explain.

Author Response

Reviewer 1:

Thank you very much for the useful suggestions. We accepted all of the suggestions and we are sure that this will improve the quality and contribute to a better understanding of the paper.

- The paper "Intelligent Novel IMF D-SWARA – Rough MARCOS Algorithm for Improvement of Performance for Construction of Road Infrastructure" is well written manuscript with a newly proposed integrated MCDM methodology.

- The authors are very familiar with the topics covered in the paper. It is well written with all the necessary elements. It is easy to read because is synthesized in a good manner.

Some of the advantages of the paper are:

- Clearly abstract with all core elements.

- The overall quality of the manuscript.

- The authors have ensured well paper structure with a clear explanation. In the introduction, they show the significance of the field first, and after that clearly aims and contributions. A literature review is very extensive and it is combined text with a tabular review which increases the quality of this section. MCDM problem is well defined with an explanation of basic elements. The methodology is novel and appropriate for solving any MCDM problem. Extensive sensitivity and comparative analysis is one more proof of excellent work.

- Developing of novel methodology with clear contributions

- Well description of variants and used criteria.

The paper has great potential and can be accepted after the following minor corrections:

Comment 1: Add the structure of the paper as the last paragraph in the introduction.

Response to comment 1: The structure of the paper has been changed and accordingly written the last paragraph of the introduction.

Comment 2: Some similar papers should be discussed in the manuscript as follows: (i) An analytics approach to decision alternative prioritization for zero-emission zone logistics (ii) Prioritizing Transport Planning Strategies for Freight Companies Towards Zero Carbon Emission Using Ordinal Priority Approach

Response to comment 2: Both references have been added.

Comment 3: In Table 8 you show an example of evaluating the performance of pavers for Expert 1. It will be useful if you add in appendix evaluation by the rest experts (E2, E3, and E4).

Response to comment 3: Thank you for your suggestion. We have added three Tables in Appendix.

Comment 4: Why did you create 30 scenarios in validation analysis? Please explain.

Response to comment 4: The following has been added in the paper: We have decided to create 30 scenarios for reason that these three criteria have the largest influence on the decision-making process with the assumption that other criteria haven’t influenced alternative ranking.

Reviewer 2 Report

1) The introduction section is poorly written. Specifically sections 1.1 and 1.2. Please improve writing to clearly understand the objectives and methodology/contributions. In respect to objectives: the research gap, research questions and goals are not clear. 

2) Literature review. Some lines are written in a very awkward way (lines 113 and 136 for example). You should consider the following reference to complement Table 1. 

Ruiz, A.; Guevara, J. Sustainable Decision-Making in Road Development: Analysis of Road Preservation Policies. Sustainability 202012, 872. https://doi.org/10.3390/su12030872

3) The problem definition section is poorly written. The whole section could be summarized as a Table. Please improve. 

4) The paper is lacking a proper "Research Method" section in which authors can describe what sources of information were used, the data that were employed, and a basic explanation of how data were analyzed. 

5) After the Research Method section, the authors should provide a "Results" and "Discussion/Analysis" sections. Results should be independent from "Discussion/Analysis". This facilitates readers' understanding of the study. 

6) Section 4 should be included into "Research Method" or should be sent to an appendix. 

7) Section 5 should be improved as it currently presents "Results" and "Discussion/Analysis" at the same time. It is very difficult to understand all the tables as it is not clear what different weights, categories, and multiple other variables mean.

8) The title is not clear and reads very awkward. Please improve. 

Author Response

Reviewer 2:

Thank you very much for the useful suggestions. We accepted all of the suggestions and we are sure that this will improve the quality and contribute to a better understanding of the paper.

Comment 1: The introduction section is poorly written. Specifically sections 1.1 and 1.2. Please improve writing to clearly understand the objectives and methodology/contributions. In respect to objectives: the research gap, research questions and goals are not clear.

Response to comment 1: Introduction section has been improved.

Comment 2: Literature review. Some lines are written in a very awkward way (lines 113 and 136 for example). You should consider the following reference to complement Table 1.

Ruiz, A.; Guevara, J. Sustainable Decision-Making in Road Development: Analysis of Road Preservation Policies. Sustainability 2020, 12, 872. https://doi.org/10.3390/su12030872

Response to comment 2: The whole second section has been corrected from mentioned aspect. Also, suggested reference has been added to Table 1.

Comment 3: The problem definition section is poorly written. The whole section could be summarized as a Table. Please improve.

Response to comment 3: We have created two new Tables according to your suggestion and it is part of new section 3. Methodology.

Comment 4: The paper is lacking a proper "Research Method" section in which authors can describe what sources of information were used, the data that were employed, and a basic explanation of how data were analyzed.

Response to comment 4: Thank you for your suggestion. We have added 3. Methodology section with necessary elements.

Comment 5: After the Research Method section, the authors should provide a "Results" and "Discussion/Analysis" sections. Results should be independent from "Discussion/Analysis". This facilitates readers' understanding of the study.

Response to comment 5: Done. Now is the fourth section Results.

Comment 6: Section 4 should be included into "Research Method" or should be sent to an appendix. 

Response to comment 6: The former section 4 now is part of section Methodology.

Comment 7: Section 5 should be improved as it currently presents "Results" and "Discussion/Analysis" at the same time. It is very difficult to understand all the tables as it is not clear what different weights, categories, and multiple other variables mean.

Response to comment 7: We have adopted your suggestion. Now Results are the fourth section, while the fifth is 5. Sensitivity, comparative analysis and discussion.

Comment 8: The title is not clear and reads very awkward. Please improve.

Response to comment 8: We have changed title of the paper. Now is Intelligent Novel IMF D-SWARA – Rough MARCOS Algorithm for Selection Construction Machinery for Sustainable Construction of Road Infrastructure.

Round 2

Reviewer 1 Report

All issues have been successfully addressed by authors.

Reviewer 2 Report

I believe the authors have responded to my comments.